# EU-ETS emergency reserve price curbs coal use and shields consumers during natural gas price shocks

Antonio M. Bento[1,2,3,7], Nicolas Koch [4,5,7] ✉ & Zissis E. Marmarelis [1,6,7]

Recurring surges in natural gas prices strain climate policy by raising electricity prices, inducing gas-to-coal switching, prompting discretionary interventions, and fueling pressure to weaken decarbonization. We develop an empirical framework that quantifies how gas price spikes compromise climate policy and provides a toolkit to assess emissions trading system responses based on environmental effectiveness and their capacity to limit high electricity prices. Exploiting the gas price shock following Russia's invasion of Ukraine, we estimate the impacts of gas prices on coal generation, $CO_2$ emissions, and electricity prices across 13 EU countries using hourly electricity market data. We find that the EU's gas price cap has limited effectiveness and instead propose a resilient rule-based emergency mechanism within the EU Emissions Trading System (EU ETS). A modest auction reserve price, automatically triggered when gas prices exceed historical benchmarks, can protect consumers while preserving decarbonization incentives and limiting the need for ad hoc interventions.

Episodes of extraordinarily high natural gas prices in international energy markets can compromise the effectiveness of climate policy and the pace of decarbonization, prompting the switch from gas to coal in the generation of electricity[1]. Higher natural gas prices can also lead to increases in wholesale electricity prices, impacting consumers and the rest of the economy[2]. Due to its dependence on Russian gas, the European energy system faces unique challenges[3]. The unexpected spike in natural gas prices following the Russian invasion of Ukraine in February 2022 underscores such challenges. In a year, gas prices increased by nearly 600%—from 18 euros per megawatt-hour (EUR/MWh) in March 2021 to 124 EUR/MWh in March 2022, achieving record high levels of about 300 EUR/MWh by September 2022 (Fig. 1 Panel A). In this period, coal prices also surged by nearly 500%, reflecting gas-to-coal substitution (Fig. 1 Panel C). Despite an almost doubling of carbon price levels under the EU Emissions Trading System (EU ETS) from 45 EUR/tonne to 75 EUR/tonne (Fig. 1 Panel D), coal power generation rose by 42% EU-wide (Fig. 1, Panel B). Wholesale electricity prices also

increased steadily from about 82 EUR/MWh to more than 200 EUR/MWh (Fig. 1, Panel E). While since September 2022 natural gas prices have declined (Fig. 1, Panel A), over the course of 2025, prices for gas futures contracts with delivery in the year 2026 are still almost 100% higher than pre-crisis levels, suggesting a new normal of higher gas prices. At the same time, future increases in the global demand for natural gas, as well as geopolitical challenges, could well revive price volatility and trigger renewed price spikes, as seen in early March 2026 when gas prices roughly doubled within a week to about 66 EUR/MWh following the escalation of the Iran conflict.

In response to the challenges that higher natural gas prices pose in relation to electricity prices and carbon emissions, some European nations have called for a faster transition away from fossil fuels through aggressive investments in carbon-free technologies to reduce Europe's exposure to natural gas markets[4]. Others, like Poland, have urged a delay in stricter climate action amid the crisis. The EU has implemented a uniform cap on natural gas prices at 180 EUR/MWh in

[1]Sol Price School of Public Policy, University of Southern California, Los Angeles, CA, USA. [2]Department of Economics, University of Southern California, Los Angeles, CA, USA. [3]National Bureau of Economic Research, Cambridge, MA, USA. [4]Potsdam Institute for Climate Impact Research (PIK), Berlin, Germany. [5]IZA Institute of Labor Economics, Bonn, Germany. [6]Environment and Society Centre, Chatham House, London, UK. [7]These authors contributed equally: Antonio M. Bento, Nicolas Koch, Zissis E. Marmarelis. ✉e-mail: nicolas.koch@pik-potsdam.de

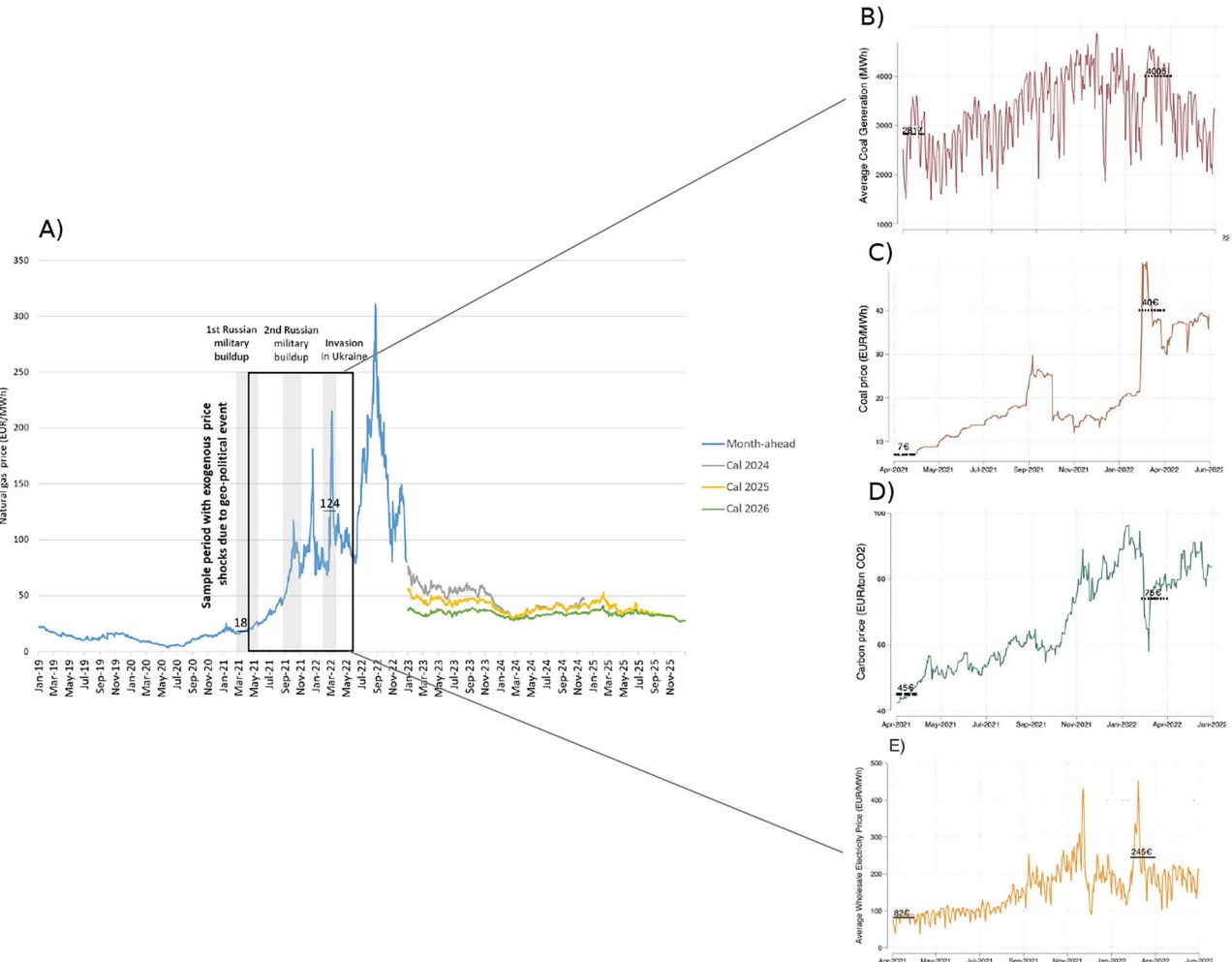

**Fig. 1 | The spike in natural gas prices in the wake of the Russian invasion of Ukraine and the resultant coal generation, coal prices, carbon prices, and wholesale electricity prices in the EU.** The box represents our sample period with plausibly exogenous spikes in natural gas prices triggered by geopolitical events highlighted by gray areas. Each panel highlights average price levels at the beginning and height of crisis, from April 1 2021 to May 30 2022. From the large spikes in natural gas prices one can clearly see the resulting rises in coal generation, carbon price, coal price, and wholesale electricity price. Policymakers thus face two different challenges–mitigating emissions from increased coal generation and mitigating the widespread consequences from increased electricity prices. **A** Reference prices for natural gas in Europe based on month-ahead futures for natural gas (blue line) and natural gas futures contract with delivery in 2024, 2025, and 2026 (gray, yellow, and green lines) traded at the TTF hub. Prices are interpolated for non-market days. **B** Average coal generation for our sample of 14 EU countries averaged at a daily level during our sample period. **C** Coal price based on month-ahead futures (ARA) at the daily level. **D** Carbon price based on futures for EU allowances (EUAs) in the EU Emission Trading Scheme (EU ETS) traded at ICE ECX at the daily level during our sample period. **E** Average wholesale electricity price for our sample of countries averaged at a daily level during our sample period.

late 2022, and considered various electricity price controls starting in October 2023[5–7]. These interventions have come under scrutiny as critics are unsure if they provide a clear solution to the soaring prices of natural gas and its impacts on consumers[8,9]. In principle, countries that have implemented an ETS to regulate greenhouse gas emissions have the potential to effectively address the challenges that high gas prices pose within the design of the ETS. However, in practise, no existing ETS has a direct response mechanism that could alleviate the effects of international energy price crises by automatically triggering an emergency mechanism in the auctions of carbon permits. Such a rule-based mechanism could apply an emergency auction reserve price that would be set to contain the gas-to-coal switch. In addition, the revenues generated through this auction reserve price could be used to provide relief to consumers.

Three largely disconnected strands of literature are relevant for designing policies that mitigate the effects of unexpected natural gas price spikes on decarbonization and electricity prices. First, a substantial body of work examines the importance of fuel switching, particularly gas-to-coal switching, in energy transitions[10–15], highlighting that high natural gas prices induce coal use and associated $CO_2$ emissions. Second, another literature analyses the pass-through of fuel costs to wholesale electricity prices[16], typically in country-specific contexts, including the United States[17], Spain[16], Germany[18] and Nordic countries[19], suggesting that high natural gas prices translate to electricity price increases that can burden consumers. Third, a separate line of research studies long-term stabilisation mechanisms in emissions trading systems—such as price floors, ceilings, and collars—that aim to reduce inter-temporal volatility in abatement costs[20]. Despite their relevance to a common policy challenge, these three strands have not been systematically integrated. As a result, we lack a unified framework for assessing how an emissions trading system can be made resilient to short-lived but extreme episodes of natural gas price spikes. Our contribution is such a unified framework that lies at the intersection of these literatures. It allows us to conceptualise and compare alternative policy options that aim to simultaneously protect consumers against high energy prices without compromising

decarbonization goals. Identifying robust emergency response mechanisms is important because political economy pressures, stemming from commitment challenges and distributional concerns, intensify during periods of energy price crises. Absent evidence-based guidance for predictable, rule-based response mechanisms, policymakers revert to ad hoc interventions that risk distorting incentives and weakening long-term climate policy commitments.

In this paper, we propose an emergency response mechanism within the existing emissions trading architecture to enhance the resilience of climate policy to energy crises. We compare its effects with gas price caps, currently the most widely used policy to address unexpectedly high energy prices. We develop an empirical framework that provides the analytical toolkit to quantify how excessive natural gas price spikes compromise climate policy and to evaluate different policy responses based on their environmental effectiveness and capacity to address wholesale electricity prices. We apply this framework to the 2022 energy crisis following Russia's invasion of Ukraine. First, we exploit the unexpected spike in natural-gas prices around the Russian invasion of Ukraine in February 2022 to estimate its causal effects on coal generation and carbon emissions across 13 EU countries that still rely on both fuels. Specifically, we employ regression techniques using hourly electricity market data. Second, combined with estimates for the impacts of natural gas prices on wholesale electricity prices, we examine the emissions and wholesale price effects of the 180 EUR/MWh natural gas price cap already implemented by the EU. Third, we propose an alternative emergency response mechanism within the EU ETS and contrast its performance against the EU gas price cap. This emergency response mechanism would build on the existing framework of the Market Stability Reserve, which already enables rule-based interventions in the auctioning of emission permits. It would be triggered automatically according to a predefined rule whenever gas prices reach levels substantially above their historical trends. When activated, permits would be auctioned only if bids exceed a predefined minimum–the emergency reserve price. This reserve price would be set to keep the relative price of gas to coal constant at a defined historical reference level. The proposed emergency mechanism is designed as a "second-best" policy that preserves the political and dynamic integrity of the ETS under conditions of market stress, and that constrains policymakers from resorting to ad hoc interventions.

Compared to the existing gas price cap, the proposed mechanism offers some advantages. First, a small increase in the price of carbon is relatively more effective in reducing carbon emissions than a cap on the price of gas. Two underlying channels of adjustment drive this difference. Both instruments operate through a substitution effect, limiting reshuffling away from gas toward coal. However, a gas price cap effectively subsidises the price of wholesale electricity, leading to an increase in demand and resulting emissions. We term this effect the output effect. In contrast, a higher carbon price induced by the proposed emergency reserve price contracts the electricity demand and reduces carbon emissions by raising the price of electricity. Only the proposed emergency mechanism allows the substitution and output effects to reinforce each other. Second, the emergency response mechanism has the advantage of generating revenues, which can be used to alleviate the consequences of higher electricity prices. Overall, we find that a modest emergency levy of 12.18 EUR/tonne on top of the regular carbon price in the EU ETS could serve as an effective response mechanism to reduce excess emissions from gas price spikes. Such a mechanism would generate greater relief to consumers than a natural gas price cap.

## Results

Figure 2 shows how the surge in natural gas prices during the 2021/2022 crisis affected coal-related $CO_2$ emissions and the pace of power sector decarbonization (panels A-D), as well as wholesale electricity prices

(panels E-G). Our focus is on the 13 EU countries that still rely on coal and gas for electricity production. We analyse the period from April 2021 to May 2022 to isolate the effect of gas price spikes triggered by exogenous geopolitical events (Fig. 1). The restriction to this time period limits confounding from policy responses and longer-term changes in generation capacity (see Supplementary Discussions 1.1 and 1.2).

### $CO_2$ emission increase from natural gas price surge

Figure 2 panels A and B quantify the excess coal generation and the excess $CO_2$ emissions from natural gas to coal switching in the generation of electricity induced by the unexpected, plausibly exogenous surge in the price of natural gas around the time of the Russian invasion of Ukraine in the period between April 2021 and May 2022. Excess coal generation and excess $CO_2$ emissions are presented relative to a counterfactual scenario, in which the relative price of natural gas to coal – subsequently referred to as the relative gas price – remains at the low pre-crisis price level (34% lower than in the crisis period). For instance, for Germany we calculate that annual $CO_2$ emissions are about 43,572 kt or 27% above this counterfactual scenario. The overall increase in emissions of 19% or 73,587 kt in all our sample countries provides clear evidence that the episode of extraordinarily high relative gas prices in the aftermath of the Russian invasion of Ukraine compromised the pace of decarbonization of the electricity sector in the EU.

Quantified excess coal generation and excess $CO_2$ emissions are determined by the product of the responsiveness of coal-fired power generation to the relative gas prices shown in Fig. 2D and the quantity of coal-fired electricity generated shown in Fig. 2F (see "Methods"). Our estimates of the price responsiveness of coal generation (see Methods for the underlying regression techniques) suggest large heterogeneity across the 13 EU countries in the magnitude of the gas-to-coal switching occurring in response to high relative gas prices. We find the greatest effects for Italy (0.75) and Spain (0.71). The estimate of 0.75 for Italy suggests that the generation of coal increases by 75% if the relative price of gas increases by one unit. The effects are more moderate in Finland, Germany, and Denmark, while they are much less pronounced in Eastern and Southeastern EU countries. A high relative price responsiveness of coal generation translates into a high relative increase in coal generation and $CO_2$ emissions. This is why we observe in Fig. 2B the largest relative emission increase induced by the rise in the price of natural gas in Italy and Spain (50% and 48%). However, the absolute emission increase in these two countries is limited. This is because the total amount of coal-fired power generation drives the absolute emission effect and the dependence on coal is relatively low in Italy and Spain (Fig. 2F). In contrast, more coal-dependent countries such as Germany, Poland, Czechia, and Bulgaria show high absolute emission increases–despite their medium to small responsiveness to relative gas prices.

### Electricity price increase from natural gas price surge

In addition to the increasing emissions resulting from the substitution from natural gas to coal power generation, high natural gas prices also pass through to higher wholesale electricity prices. Fig. 2E quantifies the excess wholesale electricity price increases induced by the 2021/2022 natural gas price surge relative to the counterfactual scenario, in which the relative gas price remains at the low pre-crisis price level. We find the greatest effects for the southern EU countries, where Greece stands out with an increase in electricity prices of 131 €/MWh (245%). The price increase is lowest in Northern Europe, such as 59 €/MWh (120%) in Finland.

The quantified excess electricity prices are determined by the pass-through rate of natural gas prices to wholesale electricity prices (Fig. 2F) and the level of wholesale electricity prices (Fig. 2G). Our estimates of the pass-through rates (see Methods for discussion of underlying regression techniques) again suggest important heterogeneity in the extent to which high natural gas prices trigger higher

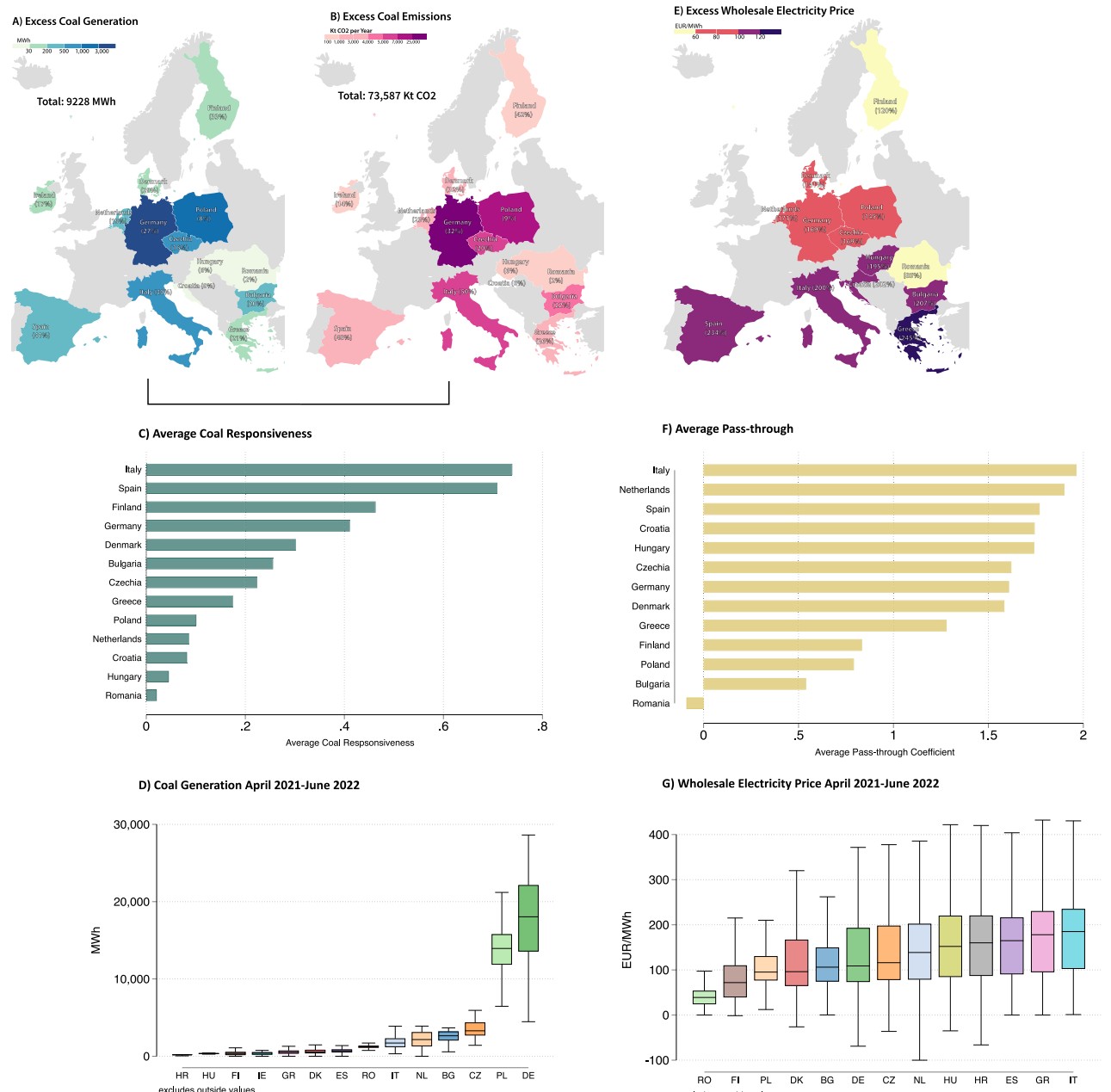

**Fig. 2 | Excess emissions and electricity price from natural gas price surge.** The estimated responsiveness of coal generation to the relative price of natural gas to coal and pass-through of natural gas to wholesale electricity price allow us to quantify the excess $CO_2$ emissions and wholesale electricity price across EU countries from April 2021 to June 2022. Maps in Panel A, B and E were created using visualisation tools by Datawrapper. **A** Excess coal and lignite generation reflects the additional coal generation above the predicted coal generation under "non-crisis price environment." This is calculated by subtracting the estimated amount of coal generated under the pre-crisis (January-March 2021) scenario with lower relative prices (34% lower at 1.25 as opposed to 1.88; Supplementary Fig 14 & 15 use alternative baselines) from the actual observed coal generation (panel D). The relative percentage increase is compared to this pre-crisis period. **B** Excess carbon emissions generated from excess coal generation (panel A), using standard lignite and hard coal emission factors (see Methods). **C** Average coal responsiveness estimates per country. **D** Average coal generation per country, with vertical box plots

capturing the ± 1.5 × IQR (the inter-quartile range) with bounds at the 25th percentile (Q1) and 75th percentile (Q3). The center line of the box represents the median (50th percentile). Whiskers extend to the smallest and largest values within 1.5 times IQR of the lower and upper quartiles, respectively. The sample size is: Bulgaria 10,223, Czechia 10,224, Germany 10,224, Denmark 10,223, Spain 10,193, Finland 8882, Greece 10,213, Croatia 7,981, Hungary 10,210, Italy 10,224, Ireland 9974, Netherlands 10,206, Poland 10,224, Romania 10,173. Supplementary Fig. 17 excludes the outliers Germany and Poland. **E** Excess wholesale price reflects the additional price of electricity above the predicted price under a "non-crisis price environment" with low gas prices (January-March 2021). This is calculated by subtracting the estimated wholesale electricity price predicted under the non-crisis scenario using the country pass-through estimates (panel F), from the observed wholesale price (panel G). **F** Average price pass-through estimates per country. **G** Average wholesale electricity price per country, with vertical box plots defined as in Panel D. The sample size for each country is 10,224 observation hours.

electricity prices across the 13 EU countries. We find the greatest effects for Italy (1.96) and the Netherlands (1.9) and the least effects for Poland (0.79) and Bulgaria (0.54). The estimate of 1.96 for Italy suggests that an increase of 1 EUR/MWh in the natural gas price results in

an increase of 1.96 EUR/MWh in the wholesale electricity price. Because the heterogeneity in the electricity price levels is limited (Fig. 2G), it is mainly the pass-through rate that drives the relative and absolute increase in wholesale electricity prices observed in Fig. 2E.

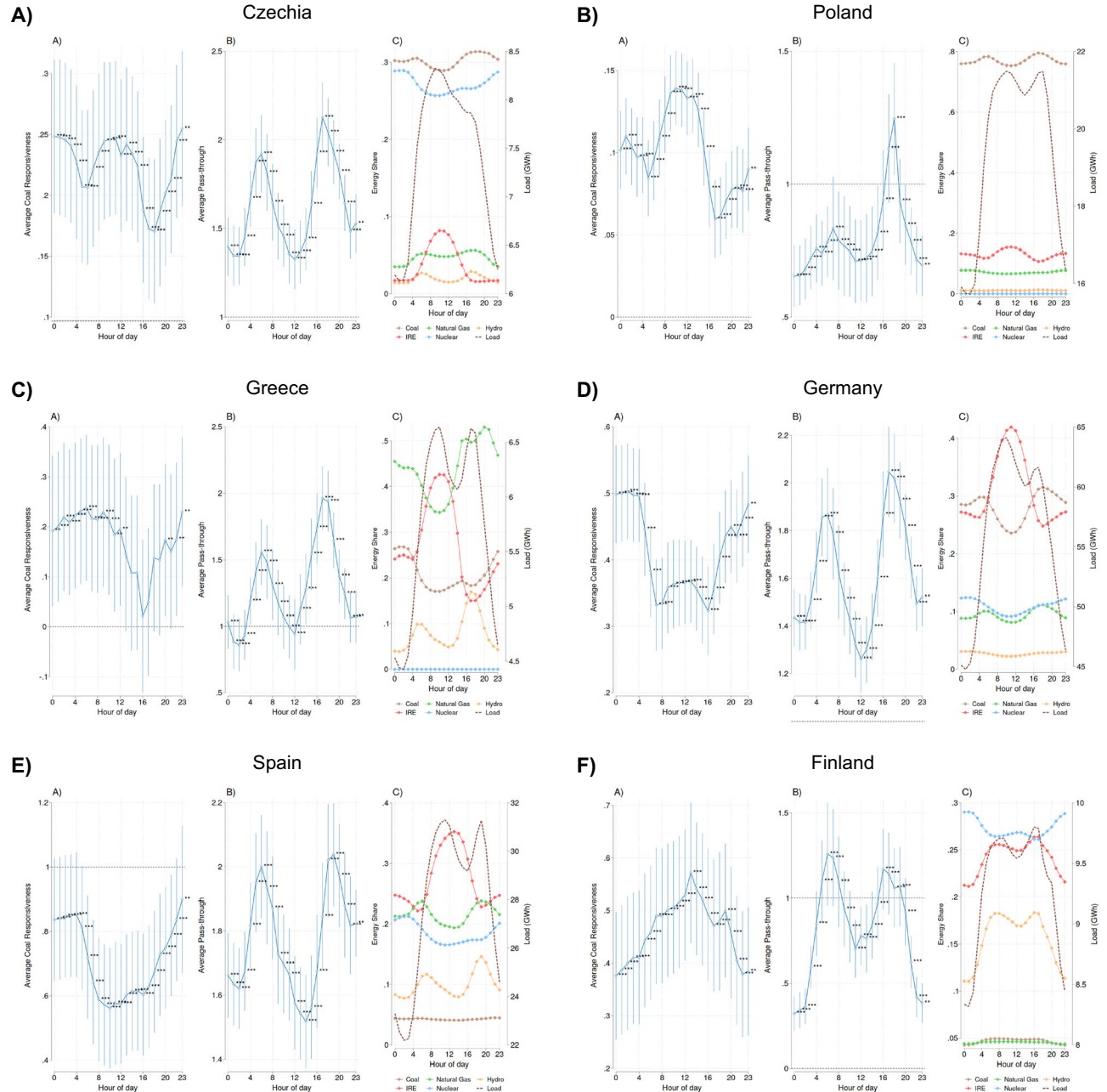

**Fig. 3 | Channels of adjustment suggested by hourly coal responsiveness, natural gas price pass-through, and energy shares for six representative countries. A–F** Estimated hourly coal responsiveness coefficients (left panels), estimated pass-through coefficients by hour (middle panels) and the average share of different generation technologies used by hour (right panels) for the following countries: Czechia, Poland, Greece, Germany, Finland and Spain, using our main specifications for each country (Supplementary Table 28 & 29) while including an hourly interaction term with our regressor of interest. These countries were selected to highlight the varying nature of the interrelationship between coal responsiveness and price responsiveness, while also providing geographic diversity. For each country, dots in the left and middle panels (center of the vertical bars) represent the estimated hourly coal responsiveness or pass-through coefficients

respectively, while the vertical error bar lines capture 95% confidence intervals ($\pm 1.96$ times the standard error of each point estimate). The statistical test used is a two-sided t-test. The coefficients that are statistically distinct from either 0 or 1 at the 1% or 5% significance level are indicated with three stars (***) or two stars (**), respectively. In the right panel, "intermittent renewable" includes solar, wind, and hydro-run-of-river. The dashed line displays the average hourly load. The sample size for each country for estimated coal responsiveness (left panels) is 10,224 observations hours in Czechia, Poland, and Germany, 10,213 in Greece, and 8882 in Finland. The sample size for estimated pass-through coefficients (middle panels) for each country is 10,224 for all countries. For the remaining country-specific figures see Panels A-C of Supplementary Fig. 20–26.

## Heterogeneity in the adjustment

In Figure 3, we decompose the channels of adjustment behind the average effects for the gas-to-coal substitution and the electricity price in response to the higher relative natural gas price by showing hourly estimates for the responsiveness of coal generation (left panel) and the responsiveness of electricity price measured by pass-through rates

(middle panel) for six selected countries with different electricity generation mixes (right panel). It shows clear hourly patterns in the two responsiveness estimates. The electricity load is one driver of these temporal patterns. In times of high load during the morning and evening peaks, we generally observe a higher responsiveness of either coal generation (i.e., more gas-to-coal switching) or electricity prices

(i.e., a higher pass-through of natural gas prices). The mix of electricity generation technologies is a second driver of temporal heterogeneity. For example, in times with abundant intermittent renewables (IRE), we generally observe a lower response of coal generation to the relative price of gas. Taken together, three different cases of adjustment can be identified from the temporal response patterns.

First, for coal-dependent countries, such as Poland and Czechia, we show a negative correlation between coal generation responsiveness and electricity price responsiveness. In particular, during hours of high electricity load (shown as dashed line in the right panel), we observe a high coal generation responsiveness but a lower electricity price responsiveness. This suggests that these two Eastern European countries use their coal capacity to balance out fluctuations in gas prices, and the available coal capacity keeps electricity prices from rising. A similar adjustment mechanism can be observed in Greece, where, however, coal capacity is not sufficient to prevent electricity price increases in the evening peak.

Second, for countries with a high share of IRE but also nuclear energy, namely Finland and Spain, we instead find a positive correlation between the coal generation responsiveness and the electricity price responsiveness. During periods of high electricity load, these two countries do not have sufficient coal capacity that could be used to replace more expensive gas generation, and thus the higher generation costs must be passed through to higher electricity prices.

Finally, Germany is an example of a country with a relatively high share of IRE but still a high share of coal. With a correlation between the coal generation responsiveness and the electricity price responsiveness close to zero, the country falls between the two cases with a clear positive and negative correlation, suggesting that in times of high relative gas prices, it can use gas-to-coal switching to prevent some of the electricity price increase but not to a larger extent than in Poland and Czechia. It is worth noting that coal-fired plants generally lack the fast-ramping capabilities of gas-fired units that make the latter particularly well suited to balancing intermittent renewable generation, which can constrain the extent of fuel switching.

## Trade-off between gas-to-coal switch and higher electricity price

The pattern with trade-offs between switching to more coal generation or paying higher electricity prices discussed for the six selected countries generalises across all our sample countries (see Supplementary Figs. 20–26). To show this, Fig. 4A presents a relative responsiveness score that is based on the negated correlation coefficient between the hourly estimates for the coal responsiveness and for the price pass-through (see Methods). The higher the score, the more negatively correlated the response of coal and the price pass-through. In addition to Czechia, Poland, and Greece, all other Eastern European countries but also Italy and the Netherlands are among the group of countries with a high relative responsiveness score. In most cases, these countries rely on coal to balance out high natural gas prices and keep electricity prices from rising. However, in some countries such as Italy (Supplementary Fig. 24) and the Netherlands (Supplementary Fig. 25) and in some periods of the day (such as late afternoon in Greece, but also Czechia and Poland), the high responsiveness score reflects cases with rising electricity prices because coal generation does not respond.

In contrast, a second group of countries is characterised by a negative score (i.e. a positive correlation between the coal responsiveness and the price pass-through), which suggests that the switch from gas to coal generation in these countries is insufficient to stop electricity prices from increasing or that these country use some alternative generation technology that prevents gas-to-coal switching. This group includes Finland and Spain, but also Croatia and Denmark, i.e., those countries that have a high share of decarbonised electricity and a relatively low coal capacity. Figure 4B, C suggest that the electricity mix is a determinant of the trade-off countries face when dealing with higher relative natural gas prices. A higher reliance on coal

generation is correlated with a higher responsiveness score and a higher share of decarbonised energy is correlated with a lower score (specifically, high shares of wind and hydro energy are correlated with a lower score; Supplementary Fig. 30).

## Effectiveness of EU gas price cap is limited

In response to the 2021/2022 natural gas price surge, the EU implemented a price cap, aiming to limit episodes of excessive gas prices[22]. The so-called Market Correction Mechanism is automatically activated if the month-ahead price for natural gas on the Title Transfer Facility (TTF) trading platform exceeds 180 EUR/MWh for three working days. The second market event triggering the mechanism is when the month-ahead TTF price is 35 EUR/MWh higher than a global reference price for LNG for the same three working days. Once activated, the cap is valid for at least 20 working days. It would be automatically deactivated if gas prices are below 180EUR/MWh for the last three consecutive working days. Even though the EU-wide mechanism was only applied from February 2023 when the crisis was already subsiding, it remains relevant to understand how it would have worked if it had already been active in 2022 when natural gas prices still soared. Figure 5A shows results for this policy counterfactual. It indicates that the emissions savings from a natural gas price cap in 2022 would have been low in most countries except Germany. For the EU in total, we estimate minor $CO_2$ emission savings of 6.867 kt $CO_2$. The environmental effectiveness is low despite the fact that the 180 EUR/MWh threshold would have been triggered 115 times in the year, as 2022 stands out as the period with record natural gas prices hitting an all-time high of more than 300 EUR/MWh (see Fig. 1 and Supplementary Fig. 14). Figure 5B highlights the channels exploited by the natural gas price cap and underscores why its effectiveness is limited. It shows two opposing effects of the cap. First, there is an emission-reducing substitution effect. By capping natural gas prices, less gas-fired power generation is substituted by coal-fired generation, and this results in lower $CO_2$ emissions. However, there is a second, emissions-increasing output effect. The lower gas prices imposed by the cap are passed through to lower wholesale electricity prices, which increases electricity demand and, thus, increases $CO_2$ emissions. Figure 5C shows the induced electricity price decrease that causes this output effect.

## Small carbon price increase can effectively reduce excess emissions

Strengthening the carbon price in the existing EU Emissions Trading System (EU ETS) is an alternative policy option that could more effectively address excess $CO_2$ emissions. A higher price on carbon makes coal effectively more expensive than natural gas, and thus works in the same direction as a price cap on natural gas which makes natural gas cheaper. We calculate that the additional price of carbon that would generate the same substitution effect that the cap on the price of natural gas yields amounts to 12.18 EUR/tonne (see Methods). In comparison, the average carbon price in the EU ETS in 2022 was 80 EUR/tonne. Figure 5A shows that the cap-equivalent carbon price would deliver more substantial $CO_2$ emission reductions because, in sharp contrast to the natural gas price cap, there is a negative output effect (Fig. 5B). This is because electricity prices would increase to some extent (Fig. 5C), which reduces electricity demand. For the EU in total, we estimate that the 12.18 EUR/tonne carbon price would reduce $CO_2$ emissions by 14,048 kt of $CO_2$, which is a reduction nearly twice that of the gas price cap.

## Consumer relief from gas price cap vs. carbon price with revenue recycling

Finally, Fig. 5D compares the financial burden and relief from the gas price cap and the equivalent carbon price. The addition of a carbon price of 12.18 EUR/tonne imposes higher electricity costs on

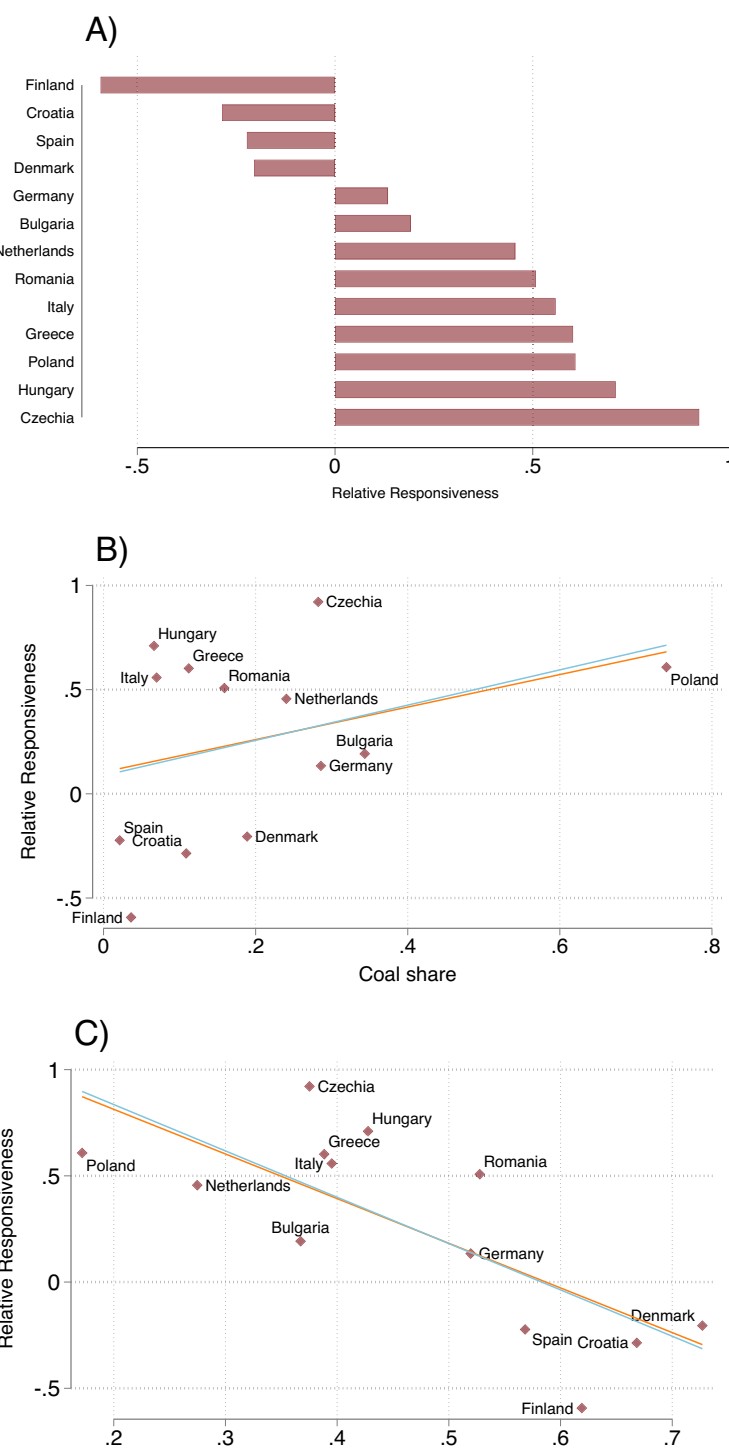

**Fig. 4 | Relative responsiveness index reflecting trade-off between gas-to-coal switch and higher electricity price.** For each country separately, a relative responsiveness score is calculated based on the hourly estimates of each responsiveness. **A** Score of relative responsiveness for all countries. This is defined as the negated Pearson correlation coefficient between the 24 hourly estimates of coal responsiveness and price pass-through (as depicted in Fig. 3A, B for a subset of countries; see Methods for more information on the index calculation). The higher the relative responsiveness score, the more negatively correlated coal responsiveness and natural gas price pass-through to electricity are. As such, it indicates that this country is reliant on coal to balance out the fluctuations in natural gas prices and keep electricity prices from rising. This is in contrast to a negative score, which suggests that coal generation is insufficient to stop the electricity price from

increasing. **B** Correlation between relative responsiveness and share of coal in electricity production. A higher coal usage is correlated with a higher score. **C** Correlation between relative responsiveness and share of decarbonised electricity production (solar, wind, hydro, and nuclear energy).A higher decarbonised energy share is correlated with a lower score.Breaking down the decarbonised share, a higher hydro or wind share is correlated with a lower score, while nuclear and solar have a slight positive correlation (Supplementary Fig. 30). Scatter plots include a fitted line of a linear regression estimated via OLS (blue line), and a fitted line estimated using the robust MM-estimator (orange line). The robust MM-estimator is preferred when the data has outliers and likely-influential but non-representative points[21].

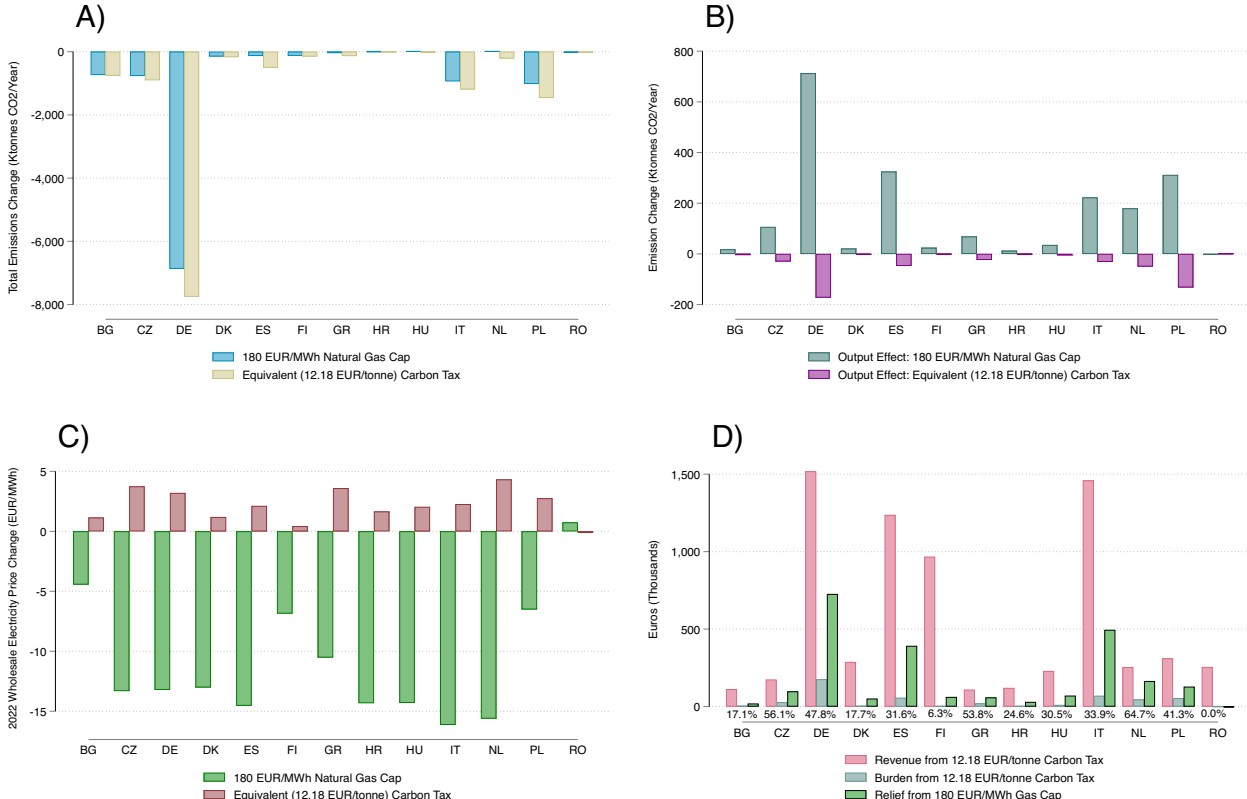

**Fig. 5 | Policy counterfactual analysis comparing the gas price cap (180 EUR/MWh) and the substitution-equivalent carbon price (12.18 EUR/tonne) over the period January 2022-December 2022.** Equivalent carbon price represents the additional price that would generate the same substitution effect that the cap on the price of natural gas yields during 2022. **A** Comparison of total emissions change under each policy per country, including the substitution and output effect. **B** Comparison of underlying output effect and substitution effect under each policy per country. The output effect per country is calculated using electricity price pass-through estimates and assumed elasticity of demand (-0.06), based on prior literature, while Supplementary Fig. 6 provides alternative estimations based on lower and higher assumptions. **C** Comparison of wholesale electricity price change under each policy per country using our pass-through estimates (see Methods). Pass-through under the equivalent carbon price is assumed to work via the corresponding increase in natural gas price, and we assume it remains unchanged aside from the increase in corresponding carbon price (see Supplementary Fig. 8 for changing assumption of carbon price pass-through to electricity price). **D** Comparison of the change in revenue and financial burden and relief of each policy per country. For the gas price cap, the relief that results from the policy-induced lowers electricity prices. For the equivalent carbon price, the burden from the additional carbon pass-through and the revenues that could be used for consumer relief is shown. The revenue is greater than the relief provided by the natural gas cap, as seen in the percentage value shown under each country's bars which shows the share of revenue left after covering the relief from the gas cap. All values are normalised by average load per country, while the revenue and burden are adjusted with country specific emissions factors.

consumers (blue bar), and the increase in the electricity price is more pronounced in countries with a high relative responsiveness score (see Supplementary Fig. 27). However, the 12.18 EUR/tonne carbon price also raises additional revenues (pink bar) that can be used for consumer compensation. These revenues are much larger than the aggregate burden (pink bar > blue bar), with the break-even ratio of revenue to burden being 8% on average. Thus, a recycling of revenues could fully offset the carbon cost burden and provide additional consumer relief of considerable magnitude (equal to the difference between the blue and pink bars, as seen in Supplementary Fig. 6). The natural gas price cap also provides consumer relief through lower electricity prices (green bar). However, the financial compensation from a full revenue recycling of the additional carbon price of 12.18 EUR/tonne would far exceed that from the gas price cap. For example, in Spain only 32% of the revenues would be required to deliver the same level of consumer relief as the gas price cap (as indicated by the percentage values under each country's bars). Even under a pessimistic scenario in which administrative and transaction costs associated with such recycling reduce the available revenues for compensation by half, revenue recycling would still provide greater relief than the gas price cap in almost all countries.

In sum, our analysis suggests that a moderate strengthening of the EU ETS combined with a financial compensation of consumers equal to all additional carbon revenues outperforms a gas price cap both in terms of environmental effectiveness and in providing consumer relief during the crisis.

## Discussion

Our study highlights the critical importance of understanding the relationship between natural gas price spikes, coal-fired generation, and resulting excess $CO_2$ emissions. The results confirm that the use of natural gas as a transition fuel has potential shortcomings[23,24]. In particular, we provide empirical evidence that episodes of extraordinarily high gas prices slow the pace of decarbonization by generating excess $CO_2$ emissions in electricity production. The response of coal-fired generation to spikes in gas prices is related to the share of coal in the electricity mix, the diversity of energy sources in electricity production, and how responsive a country is to natural gas prices. Countries with a mix of electricity generation that still rely relatively more on coal and are more sensitive to gas prices tend to experience the largest excess $CO_2$ emissions. Because natural gas and renewables are complementary, countries with large renewable investments remain

exposed to natural-gas price shocks. However, their increase in emissions is smaller due to limited coal capacity. Taken together, these results highlight the role of coal phase-out and energy source diversification, since coal generation can temporarily increase during times with extraordinarily high natural gas prices. Our results complement recent literature that stresses the need to promote a zero-carbon electricity sector to achieve climate goals[25,26].

European electricity systems are physically constrained and highly interconnected. Fuel-switching possibilities are shaped by transmission capacity, congestion, ramping constraints, and interconnector availability. Our empirical strategy relies on realised hourly generation and price data during the crisis period. These market outcomes already reflect physical and institutional constraints. Accordingly, the estimated fuel-switching elasticities reflect substitution that was feasible under actual system conditions. Cross-country heterogeneity in estimated responses therefore captures differences in generation portfolios, grid structure, interconnection capacity, and operational flexibility. Our analysis is thus reduced-form and equilibrium-based, rather than a simulation of unconstrained engineering dispatch.

Our findings have several policy implications. First, the effectiveness of the existing EU gas price cap is limited. Our framework helps explain the mechanisms that lead to this result. By lowering the price of gas relative to coal, the gas price cap promotes a substitution effect that leads to a reduction in emissions. However, to the extent that gas price caps lower the overall wholesale electricity price, electricity demand increases and emissions rise. This output effect partly neutralises the substitution effect, compromising the effectiveness of the gas cap. This result suggests that the gas price cap suffers from some of the same limitations noted in studies that examined the effectiveness of low carbon fuel standards[27], and Clean Energy Standards[28].

Second, a small carbon price in the existing EU emissions trading system can overcome the limitations of the gas price cap. In the 2022 crisis, a carbon price of 12.18 EUR/tonne would reduce $CO_2$ emissions more than twice as much as from a gas price cap. This carbon price can generate the same substitution effect as the gas price cap. But because wholesale electricity prices increase with a carbon price, the output effect would instead reinforce the substitution effect. This insight suggests potentially important reforms for the existing EU ETS. Our results that the EU ETS, as well as most existing emissions trading schemes, would benefit from an explicit emergency mechanism to address international energy price crises. Recent policy debates in Italy —including a reform proposal to decouple carbon costs from wholesale electricity prices in order to reduce consumer bills, and broader public calls to reconsider elements of the EU ETS to better balance decarbonization goals with energy costs—underscore the relevance of a predictable, rule-based mechanism that preserves the credibility and integrity of the ETS under conditions of market stress.

Political economy considerations provide a second-best rationale for our proposal of a rule-based emergency mechanism. In a first-best setting, temporary surges in coal use and emissions would not pose a concern, as the emissions cap in the EU ETS fixes cumulative emissions. However, this perspective overlooks two important real-world dynamics. First, high energy prices trigger policy debates about the political commitments to the EU ETS cap. Because the cap is not static but subject to periodic updates and unexpected policy revisions, the literature has long recognised that policymakers governing an ETS face a classic time-inconsistency problem with ex-post incentives to renege on ex-ante commitments[29,30]—with these pressures intensifying during periods of crisis[31,32]. The proposal of Poland and other Eastern European countries to abandon the scheduled tightening of the EU ETS cap following the onset of the war in Ukraine[33] illustrates the commitment problem that can arise during crises. Although the proposal failed to gain majority support, the political debate surrounding the long-term cap's stringency induced uncertainty among market participants. Such uncertainty has been suggested to deter investment in clean technologies, potentially undermining the dynamic efficiency of the ETS[34]. The proposed emergency mechanism for the EU ETS would act as a commitment device designed to preserve the political and dynamic integrity of the ETS under conditions of market stress. Second, the first-best perspective neglects policymakers' concerns about the political repercussion of distributional impacts associated with high energy prices during crisis. These concerns often trigger calls for relaxing climate policy or for alternative interventions that promise financial relief. Absent a readily available mechanism to address them, policymakers tend to adopt ad hoc measures that distort incentives and are limited in their effectiveness. The EU's introduction of the gas price cap in 2022 is a salient example of such an ad hoc intervention. The emergency mechanism would act as a rule-based device aimed at providing a predictable response to distributional concerns, which should constrain policymakers from resorting to ad hoc interventions.

Currently, the EU ETS lacks a dedicated emergency mechanism. However, the Market Stability Reserve (MSR) of the EU ETS already provides an institutional framework for rule-based interventions, and extending its mandate presents an administratively feasible way to integrate the proposed emergency levy into the existing governance architecture. The MSR automatically adjusts the supply of emission allowances (EUAs) according to quantity-based rules: when the number of allowances in circulation falls below a specified threshold, allowances are released from the MSR for auctioning, and when it exceeds an upper threshold, a fraction of the surplus is withdrawn. More specifically, the rule is based on the Total Number of Allowances in Circulation (TNAC). If the TNAC exceeds 833 million, 24% of the surplus is placed in the MSR. If the TNAC falls below 400 million, 100 million allowances are released. The emergency mechanism could be incorporated into the MSR via an additional price-based rule: when natural gas prices exceed a pre-specified threshold, allowances would be auctioned with a predetermined top-up. Price-based mechanisms of this type are well established in the literature[35–37] and already implemented in the UK, California, or in the U.S. Regional Greenhouse Gas Initiative—indicating that existing institutions have the administrative capacity to implement such schemes. Even within the EU ETS, Article 29a in principle already provides for a price-based emergency intervention linked to the carbon price: if EUA prices rise "more than twice" the two-year average and the increase is not justified by fundamentals, the European Commission may release additional allowances. However, this rule is discretionary and has never been used. By contrast, the emergency mechanism proposed here would be rule-based and linked to natural gas price levels.

The implementation of the emergency mechanism within an extended MSR framework requires defining three rules. The first defines the criterion for identifying excessively high natural gas prices that would trigger the mechanism. A simple option is to specify a fixed absolute price level, such as the 180 EUR/MWh threshold used for the EU gas price cap. An alternative option, following the logic of Article 29a, is to define the threshold based on a proportional increase in natural gas prices (or in the relative price of natural gas to coal) relative to a historical reference period—for instance, a rise exceeding twice the two-year average natural gas price. Once the average natural gas price over the preceding month exceeds the pre-specified threshold, the emergency mechanism is activated automatically. Upon activation, a second rule determines a reserve price that applies temporarily to all EUA auctions. Under this rule, allowances in auctions are only released when the bid price exceeds the pre-specified minimum—what we refer to as the emergency reserve price. A simple way to set this reserve price is to add a fixed top-up, such as 10 EUR per ton, to the average bid price in the previous auction round. A more flexible alternative, adopted in our policy counterfactual analysis above, is to define the top-up based on the actual carbon price increment required to keep the relative price of natural gas to coal (i.e., the gas-to-coal price ratio)

constant at a specified historical reference level—for instance, again, the two-year average relative price of natural gas to coal. The equivalent carbon price can be calculated for any combination of natural gas and coal prices (see "Methods"). A third rule determines the duration of the emergency activation. Following the design of Article 29a, we propose a fixed period of six consecutive months. After six months, the price situation would be reassessed.

From a political feasibility perspective, a challenge of the emergency mechanism is that it raises electricity prices precisely during periods of already high energy costs. However, our analysis indicates that the emergency reserve price generates revenues that exceed aggregate household costs from higher electricity prices. In our counterfactual analysis for the 2022 energy crisis, only about 8% of the revenue would have been needed to fully offset the rise in consumer costs. This demonstrates that, even after accounting for administrative costs of the scheme, the emergency mechanism has substantial potential to provide meaningful relief to households. Recent literature on public support for carbon pricing[38–40] suggests that designing a revenue recycling scheme is a necessary condition for the political acceptability of such policies. While the EU lacks a legal framework for direct financial transfers to citizens, existing EU ETS compensation funds already provide an institutional basis for a refund mechanism. Notably, the Social Climate Fund (SCF), established to offset costs from carbon pricing[41], could serve this purpose. It is financed through ETS auction revenue and it distributes these funds to EU Member States according to a formula designed to mitigate distributional tensions across the diverse EU economies. Member States then use the funds they receive conditionally on national Social Climate Plans approved by the EU, which determine how citizens are compensated. All revenues generated by the proposed emergency mechanism (net of administrative costs) could be channelled into the SCF. It would be in the self-interest of Member States to utilise the additional revenues from the emergency mechanism to provide financial support directly to households in times of crisis.

We acknowledge some limitations of the proposed emergency mechanism. First, while we focus on the potential of providing relief to consumers by recycling the revenues of the emergency mechanism, we do not account for consumer attitudes toward taxation or trust in revenue recycling. Previous studies show that public acceptance of carbon taxation depends on clear ex ante revenue use[42,43]. In the context of the COVID pandemic, state capacity to transfer funds directly to citizens has been identified as a key prerequisite for a credible compensation policy[44]. This raises broader questions about the long-term reliance on indirect redistribution mechanisms in the EU, such as the SCF or similar COVID-era assistance programs, particularly in the context of future crisis responses. Second, we also abstract from potential behavioural failures that may cause consumers to focus on higher electricity prices while underestimating the benefits of revenue recycling[45]. Third, we abstract from potential competitiveness effects of higher electricity prices for industry that may require additional compensation mechanisms. Fourth, we acknowledge that the proposed emergency mechanism would add another layer to the already complex EU ETS architecture. However, it could build exclusively on existing MSR institutions, and unintended interaction with the current framework is unlikely. This is because the MSR's quantity-based rules are designed to address longer-term structural imbalances in the EU ETS. Activation is determined by the total number of allowances in circulation, but there is a two-year gap between the year it is measured and the year of the MSR intake or release. By contrast, the proposed, price-based emergency mechanism would respond immediately to excessive gas prices and operate only for a short, predefined period, such as six months. Given this limited and temporary nature, it is unlikely that the emergency intervention would materially influence the MSR activation two years later.

Finally, our analysis focuses on short-run operational responses during a crisis episode and does not explicitly model long-run capital adjustment. Nevertheless, the proposed emergency mechanism may have implications beyond the short run. By preserving a credible and undistorted carbon price signal during periods of extreme gas price volatility, the mechanism can help maintain stable investment incentives for renewables, storage technologies, demand response, and electrification. In contrast, recurrent interventions that suppress electricity prices through gas price caps risk weakening long-run investment signals and increasing policy uncertainty. At the same time, the relevance of the mechanism is likely to evolve as the electricity system decarbonises. In the short run, when coal and gas plants remain central to marginal electricity supply, large gas price shocks can induce substantial fuel-switching toward more carbon-intensive generation. However, as investment in renewables, storage, and demand flexibility expands and the fossil share of marginal generation declines, extreme gas price movements should have a smaller impact on aggregate emissions and wholesale prices. In such a setting, the emergency mechanism would be activated less frequently. The instrument is therefore inherently transitional and state-dependent: its importance diminishes as reliance on fossil fuels declines.

## Methods
### Data and sample
The majority of our data for the econometric analysis is obtained from the Association of European Transmission System Operators for Electricity (ENTSO-E). Transmission System Operators (TSO) generally correspond to countries, with the exception of Germany and Denmark, which are split into four and two respectively. We obtain hourly data on wholesale electricity prices (EUR/MWh), electricity generation by technology (MWh), and load (MWh). Hourly energy generation is obtained for each available source in each country and includes biomass, coal, lignite, natural gas, dispatchable hydro, nuclear, oil, solar, geothermal, wind, hydro-run-of-river, waste, and other. We focus on those 14 EU countries that utilise hard coal or lignite, in addition to natural gas, in their electricity mix (Supplementary Fig. 9 and 29). This is a requirement for our empirical approach to work as we aim to quantify the substitutability between coal sources and natural gas. This leaves Bulgaria, Croatia, Czechia, Denmark, Spain, Finland, Germany, Greece, Hungary, Ireland, Italy, the Netherlands, Poland, and Romania. The island states of Malta and Cyprus do not have available electricity generation data, while Ireland is left out of the policy analysis (Figs. 4 and 5) due to missing electricity price data.

Price data of natural gas, coal, and carbon are obtained from the Intercontinental Exchange (ICE). The natural gas price data (EUR/MWh) refer to the TTF month-ahead daily futures price (the benchmark gas price of European markets). The coal price data (EUR/tonne) refer to the ARA month-ahead daily futures price. The appropriate conversion to EUR/MWh is used under the assumption of 8.14 tonnes of coal per MWh. Similarly, the carbon price, which refers to the daily futures price for EU allowances in the EU ETS, is in units of EUR/tonne $CO_2$, and converted using an EU-wide average emissions factor of 0.3 $gCO_2$ per kWh, based on the 2018–2021 average emissions intensity of EU countries. In the policy analysis, we use country-specific emission factors given each country's electricity generation mix.

Our sample comprises a total of 10,224 h spanning the time period of April 2021–May 2022. We focus on data from this time period as it reflects a period in which natural gas prices experienced massive exogenous shocks largely due to the ramping up and eventual conflict in Ukraine and can thus be treated as plausibly exogenous, before any wholesale market distortive mechanisms were put in place (see Supplementary Discussion 1). Hence, this allows us to elucidate electricity generator responses to commodity prices and the environmental impact of the natural gas price crisis. We note that the beginning of our sample period coincides with the late phase of the COVID-19

pandemic, when economic activity and natural gas demand in the EU were still depressed, exerting downward pressure on gas prices. Our analysis therefore spans two distinct crisis episodes with opposing price dynamics: a negative demand shock associated with the late-COVID period and a subsequent positive price shock following the invasion of Ukraine. Given the short time period and the long time it takes to construct new fossil fuel facilities, it is reasonable to assume that the capacity mix is virtually unchanged. While most countries have near complete datasets for this time period, Finland, Croatia, and Ireland have the most missing data, with a total non-missing dataset of 8882, 7981, and 9974 respectively. The remaining countries have fewer than 50 h of missing data (Supplementary Table 28). For the electricity price pass-through regressions, all countries have complete data (Supplementary Table 29).

## Econometric model of coal responsiveness

We use the aforementioned exogenous variation in prices to run our main regression specification separately for each country from from 1 April 2021–30 May 2022 as follows:

$$
\begin{aligned}
CoalGen_{i,t} = {} & \beta_1^i \left(\frac{Gas_p}{Coal_p}\right)_{i,t} + \beta_2^i \left(\frac{Gas_p}{Coal_p}\right)_{i,t}^2 + \beta_3^i \left(\frac{Gas_p}{Coal_p}\right)_{i,t}^3 \\
& + \beta_4^i\, IRE_{i,t} + \beta_5^i\, Load_{i,t} + \beta_6^i\, Load_{i,t}^2 \\
& + \zeta_{m,i} + \delta_{h,i} + \gamma_{w,i} + \epsilon_{i,t}
\end{aligned}
\tag{1}
$$

Our preferred specification utilises hourly electricity generation and load data, and daily commodity price data. $CoalGen_t$ refers to the log transformed generation of aggregated coal and lignite at any given hour. Our regressor of interest is the price ratio of natural gas to coal, also known as the relative price. The relative price is inclusive of carbon prices and proportional to the amount of $CO_2$ emitted by coal or natural gas (natural gas emits one third as much as coal). In doing so, we implicitly assume that endogeneity between the carbon price and fuel switching is negligible, which is supported by evidence suggesting that fuel-switching behaviour accounts for only a small share of carbon price variation[46–49] and that carbon prices are largely determined by political and institutional factors affecting allowance supply rather than by contemporaneous emissions demand[30–32]. The cubic form of the relative price is used to allow for the flexibility of generator responses to thresholds of these prices that may experience potential nonlinearities. That is, at a certain price point coal generation may exhibit changes in behaviour unexplained by a linear model, due to ramping constraints. This is in line with previous work on coal capacity factor responsiveness to relative prices in the U.S. context[12]. Alternative specifications including only the quadratic or the linear term are included (Supplementary Table 14), in which the fit is worse than the cubic form through a larger BIC statistic. At the same time, it remains highly plausible that other functional forms (e.g., fourth or fifth power) could also yield qualitatively similar results and we choose this avenue as it is most in line with the extant literature[12].

Identification is based on month-of-year, hour-of-day, and day-of-week fixed effects, included as $\zeta_m$, $\delta_h$, and $\gamma_w$, respectively. We thus rely on within-hour and within-month variation across our sample period and natural gas price shocks due to events outside of the energy sector's domain. We further control for load (flexibly) and intermittent renewable energy generation (solar, wind, and hydro-run-of-river). To address possible heteroskedasticity and serial correlation in commodity prices and coal generation, we cluster standard errors at the level of variation in the treatment variable (natural gas and coal prices), which is daily. We run further robustness checks with different variations of fixed effects, covariates, and functional form, as shown in Supplementary Tables 1–13 for all countries. We further run robustness checks with different time period samples, as shown in Supplementary Tables S16–S19. While most countries show consistent results, few are

insignificant when the sample window is shortened before the massive price spikes in 2022, suggesting that a certain relative price threshold was needed to be reached to see the substitution effect.

To validate our choice of standard errors clustering for heteroskedasticity and autocorrelation concerns, the ACF of the regression residuals for each country confirms strong intraday correlation, and the PACF shows dependence of the first few lags, in line with operational fuel-switching dynamics and inertia or ramping of coal generation (Supplementary Tables 22–23). An additional check including a lagged regressor of the relative price of the previous day (24 h before) is included, yielding virtually identical results (Supplementary Table 20), providing no evidence of delayed fuel-switching adjustments, suggesting that generation decisions respond to contemporaneous fuel prices rather than lagged price signals. An additional robustness check is performed by running the pooled regression (Supplementary Table 21) to get an average estimate for all countries, including country fixed effects and thus controlling for any EU-wide shocks. This elasticity estimate (0.21) is significant and similar to the average of the 14 country-specific estimates (0.26).

To ease interpretation, we calculate marginal effects using the delta method and present them as the focal estimate of interest. Through this, we are able to estimate the substitution effect for each country $i$. Specifically, the marginal effect is calculated as:

$$
\mu_i = 3*\beta_3^i * \overline{\left(\frac{Gas_p}{Coal_p}\right)_i^2} + 2*\beta_2^i * \overline{\left(\frac{Gas_p}{Coal_p}\right)_i} + \beta_1^i
\tag{2}
$$

This marginal effect, $\mu_i$, which represents the responsiveness of the coal generation to the relative price of gas and coal, is then multiplied by the average generation of coal and lignite of each country during the sample period (due to the log-scaling) to obtain a value that reflects the change in coal generation per unit of relative price increase. Utilising the change in relative price for the specified period, we can calculate the change in coal generation in each country at this time (Fig. 2A). Based on the standard emission factors (lignite: 1100 $gCO_2$/kWh; hard coal: 830 $gCO_2$/kWh), we obtain the induced change in $CO_2$ emissions (Fig. 2B). The overall increase across our sample of countries is calculated as the sum of excess emissions for our sample countries during this time period. We further calculate the marginal effect $\mu_i h$ at each hour of the day for each country, by including an hourly interaction term (Fig. 3).

## Econometric model of pass-through

Pass-through of natural gas prices to wholesale electricity prices in European countries is driven by whether natural gas is the marginal fuel on the merit order system at any given hour. The level of pass-through dictates the change in wholesale electricity price, which in turn influences the studied effects of each policy. Our preferred econometric model of electricity price pass-through is discussed below, based on previous work by quantifying this level of pass-through across European countries during the crisis ([50]). The model regresses hourly electricity prices on daily natural gas prices, with alternative models used to examine the robustness of our estimates (see Supplementary Tab. 29 for main estimates). Each country is estimated separately to determine the market-wide pass-through of natural gas prices to wholesale electricity prices, as well as for every hour of the day by including an hourly interaction term. Through this, we are also able to calculate the excess electricity price during our sample period, compared to the counterfactual when natural gas prices were at pre-crisis levels (Fig. 2E).

For each country $i$, we separately estimate the following regression specification:

$$p_{i,t}^{Electricity} = \beta_i^h p_{i,t}^{Gas} + \gamma_{1,i} IRE_{i,t} + \gamma_{2,i} Load_{i,t}$$
$$+ \gamma_{3,i} Load_{i,t}^2 + \delta_{m,i} + \eta_{d,i} + \zeta_{h,i} + \epsilon_{i,t} \quad (3)$$

where hourly electricity price $p_t^{Electricity}$ is regressed on daily natural gas prices $p_t^{Gas}$, exogenous controls including hourly intermittent renewable energy generation of solar, wind, and hydro-run-of-river $IntermittentRenewables_t$ (dispatchable hydro is not included, since it is endogenous), hourly load $Load_t$ and its quadratic $Load_{it}^2$, month fixed effects $\delta_m$, hour fixed effects $\zeta_h$, and day-of-week fixed effects $\eta_d$ (see Supplementary Discussion for an exception of month fixed effects regarding Greece). This is estimated for the period of April 2021–June 2022, with a robustness check of January 2022–December 2022 and January 2021–December 2022 (see Supplementary Tab. 26–27).

The coefficients of interest, $\beta^h$, that we obtain for each country $i$ are the changes in hourly $h$ wholesale electricity price (EUR/MWh) per 1 EUR/MWh increase in TTF natural gas prices. Month and hour fixed effects are a key control variable as they control for any systematic, unobservable trends over the time sample that may be correlated with gas and electricity prices (e.g. drought, planned nuclear outages). Day-of-week fixed effects $\eta_{dh}$ similarly control for any systematic, unobservable hourly differences in prices on different days of the week (e.g., weekday vs weekend). Thus, within the same month, on the same day of the week, with the same intermittent renewable energy generation and load, we are statistically comparing two otherwise identical hours, but for the difference in daily gas prices.

### Relative responsiveness index
To understand the hourly interplay between coal and natural gas usage in each country, we construct an index that relates the hourly coal responsiveness to the hourly natural gas price pass-through coefficients. Specifically, we utilise the Pearson correlation coefficient with the 24 time points for each country, calculated as the covariance of the two estimates, divided by the product of their standard deviations as:

$$Relative\ Responsiveness = -\frac{1}{n}\left(\frac{\sum(A_i - \bar{A})(B_i - \bar{B})}{\sqrt{\sum(A_i - \bar{A})^2 \sum(B_i - \bar{B})^2}}\right) \quad (4)$$

whereby A is the hourly coal responsiveness, and B is the hourly pass-through coefficient (the left and center panels of Fig 3. for a subset of countries and Panels A–C of Supplementary Fig. 20–26 for the rest), and n is the number of observations (24). Intuitively, this is negated to ensure that a more positive score is also a more responsive country to the competition between coal and natural gas. A score of 1, which reflects a perfectly negatively correlation, is interpreted either as that country being very reliant on coal to balance out the fluctuations in gas price and prevent electricity prices from rising, or that coal generation does not increase and thus wholesale prices increase. Conversely, a score of −1 can be interpreted as that country using both coal and natural gas at any given hour and the coal is insufficient in preventing prices from rising. Further, a higher relative responsiveness score can suggest that when coal generation does down (e.g., due to a policy such as the one we suggest) wholesale electricity prices are more likely to go up. On the other hand, a low score suggests that coal is not eliminating this vulnerability to higher price. In presence of a gas cap, when coal generation goes down, the price responsiveness effects are diminished.

### Policy analysis
We use our estimates of coal responsiveness, $\mu_i$, and pass-through, $\beta_i^h$, for each country, to assess the environmental and economic impact of counterfactual policies imposed on natural gas or carbon prices during 2022, as shown in Figure 3. It is important to note that in each of these policy scenarios, we assume that while the cap or tax is imposed all other prices remain the same and are not directly affecting coal or natural gas prices. We subsequently explain the calculation of the responses shown in Fig. 5.

First, each policy creates a so-called substitution effect of an emissions change from the change in coal generation in response to the relative price of natural gas to coal (Supplementary Fig 4). The effect is determined through the responsiveness of coal generation to the relative price based on our estimates for $\mu_i$ (Supplementary Tab. 28) and the coal generation of each country. For instance, a cap on the price of natural gas makes coal relatively more expensive and thus disincentivizes its usage in lieu of the alternative. A certain level of a carbon price can have the same effect, since coal is more emissions intensive and thus a 1 EUR increase in the carbon price makes coal relatively more expensive in relation to natural gas. We determine that equivalent additional carbon tax to be 12.18 EUR/tonnes, through an iterative approach. That is, the additional carbon price that would have been needed in 2022 to cause the exact same coal-to-gas switch as the natural gas price would. Specifically, the added price of carbon is found for which the relative price (inclusive of carbon price) during the year of 2022 is equivalent to the average relative price under the hypothetical natural gas cap in this period, as shown in SI Equation (1). The underlying assumptions are the assumed average emission factors of natural gas, in comparison to coal. The iterative approach entails a grid-approach of 10000 points calculating the new relative price with incremental carbon prices (from 0.1 EUR/tonne to 20.0 EUR/tonne) until it equated the relative price under the natural gas price cap in 2022.

Second, each policy induces a change in the wholesale electricity price (Fig. 5C). This is determined through the change in natural gas price multiplied by the level of pass-through of natural gas to electricity prices, $\beta_i^h$, (Supplementary Tab. 29). The pass-through of the change in price from carbon is assessed through its impact on the price of natural gas, in which we assume 0.37 EUR/ton CO$_2$ is passed through for every 1 EUR/MWh of natural gas, given its relative lower emitting nature than coal. Taking Germany (DE) as a numerical example, the natural gas cap reduces the average wholesale price of electricity for 2022 by 13.2 EUR/MWh, while the equivalent carbon tax increases it by 3.2 EUR/MWh, using the pass-through coefficient of the natural gas to electricity price of 1.61 multiplied by the change in natural gas during this period (8.2 EUR/MWh), or the change in carbon tax adjusted via the country-specific emissions factor (2.0 EUR/MWh for Germany after the conversions), respectively.

Third, through the impact on the wholesale electricity price each policy induces demand effects for electricity and thus change in emissions−the so-called output effect (Fig. 5B). Intuitively, an increase in the wholesale electricity price yields a certain reduction in emissions given that a higher price disincentives the consumption of electricity. By assuming an average short-run elasticity of demand of electricity price coherent with the extant literature of −0.06, we are able to calculate this effect for each policy, given each country's average emissions factor (the average emissions from an additional kWh generated in each country's grid). Though previous studies for this estimate vary considerably[51–58], we use a conservative estimate, with different assumptions yielding qualitatively similar results (Supplementary Fig. 6). Continuing the example of Germany, we arrive at an output effect increase of 712 MWh and −172 MWh respectively for the natural gas cap and the equivalent carbon tax, by multiplying the change in electricity wholesale price (−13.2 or 3.2 EUR/MWh from above) by the elasticity (0.06) and average load (54.96 MWh) and dividing by the average electricity price (236.1 EUR/MWh). This is then converted to emissions from generation using the country specific emissions factor (tCO$_2$/MWh). The sum of the output and the substitution effect (7580

ktonnes CO$_2$/year) reflecting the total emissions change for each policy is depicted in Fig. 5A, specifically a reduction of 6867 ktonnes CO$_2$/year for the natural gas price cap and a reduction of 7752 ktonnes CO$_2$/year for the carbon levy.

Fourth, each policy can induce a relief and burden on consumers (Fig. 5D). This impact is assessed through the change in wholesale electricity price adjusted via each country's average load and country-specific average emissions factor, to obtain units of EUR and allow for appropriate comparisons. That is, a given additional carbon levy increases the amount of revenue equivalent to the carbon levy (in units of EUR/MWh by adjusting for the average country-specific emissions factor) multiplied by average load (MWh), while similarly burden the country by an additional amount corresponding to the increase in wholesale electricity price multiplied by average load. Continuing the case of Germany, the revenue is equivalent to 12.18 times the average load (54.96 MWh) divided by the emissions factor (0.441 tCO$_2$/MWh) times 1000 for unit conversions for a total of 1518,000 EUR. The relief is similarly calculated as the average load times the price change under the gas cap times 1000 to convert units, yielding 130,000 EUR for Germany. The burden is calculated as the average load times the price change under the carbon levy times 1000 to convert units, yielding 176,000 EUR for Germany. As shown, approximately 12% of the revenue is needed to offset the burden from the increase in electricity price for Germany. On average across countries, this value is just 8%, while the relief generated from the natural gas cap is a 32% of the value of the revenue.

### Reporting summary
Further information on research design is available in the Nature Portfolio Reporting Summary linked to this article.

### Data availability
The datasets used in this study were compiled from publicly available power generation data by the European Network of Transmission System Operators for Electricity (ENTSO-E), and carbon, coal, and gas price data from the Intercontinental Exchange (ICE). The resulting dataset (in .dta format) is available in a public code repository archived on Zenodo and can be accessed via https://doi.org/10.5281/zenodo.19289451.

### Code availability
All analyses reported in this study used the statistical software Stata (v. 16.1). All codes (.do files) used for analysis and figure creation are available publicly at Zenodo[59] and can be accessed at https://doi.org/10.5281/zenodo.19289451.

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

## Acknowledgements

We thank Taraq Khan for their insightful discussions and contributions to an early version of this work.

## Author contributions

A.M.B., N.K. and Z.E.M. contributed equally to this work.

## Funding

A.M.B., N.K. and Z.E.M. declare no relevant funding. Open Access funding enabled and organized by Projekt DEAL.

## Competing interests

The authors declare no competing interests.

## Additional information

**Peer review information** : *Nature Communications* thanks Anna Creti, Felix Mormann and the other, anonymous, reviewer(s) for their contribution to the peer review of this work. A peer review file is available.

