## [Transparent Peer Review file · Nature Communications]

EU-ETS emergency reserve price curbs coal use and shields consumers during natural gas price shocks

Corresponding Author: Dr Nicolas Koch

Version 0:

Reviewer comments:

Reviewer #1

(Remarks to the Author)

This article uses the 2021-22 spike in natural gas prices in Europe, induced by the escalating crisis and eventual war in Ukraine, as a natural experiment to assess the price spike's effect on fuel-shifting from gas to coal along with attendant changes in carbon emissions and wholesale electricity prices. Using hourly electricity market data for select EU countries, the authors model the hypothetical impact that the EU's natural gas price cap (adopted after prices were already normalizing) would have had, if in force at the time. They then compare the results of this modeling exercise to a second hypothetical scenario in which they replace the price cap with an emergency levy added to ETS clearing prices so as to neutralize the price advantage coal-powered electricity generators enjoyed relative to their natural-gas-fueled competitors during the crisis. The authors claim that their proposed emergency levy, assuming revenue recycling via disbursement to ratepayers, would prove superior to the adopted price cap in terms of both lower carbon emissions and lower electricity price burdens for ratepayers.

The article addresses an important question using a relatively novel dataset and methodology; it is well written (for rare errors in grammar see below) and follows a logical structure.

That said, I cannot recommend publication unless and until several issues (some major, others less so) are addressed.

Major issues:

1. Much of the claimed superiority of the authors' proposed emergency levy relative to the EU price cap hinges on hypothesized output effects in the form of excess electricity consumption driven by suppressed electricity prices, courtesy of the price cap – compared to overall higher electricity prices in the emergency-levy scenario. But authors reach this conclusion by assuming a degree of price elasticity in electricity demand that the pertinent literature, including the sources cited by the authors themselves, does not support. Specifically, authors assume “an average short-run elasticity of demand of electricity price coherent (sic) with the extant literature of 0.15 (pp.11-12). Yet, the cited literature, including several meta studies, overwhelmingly suggests significantly lower price elasticity values. Cserekylei (EN 40), for example, finds short-run elasticity values of -.08 (residential) and -.10 (industrial) (ibid. p.8). Jin & Kim (EN 37) find even lower short-run elasticity of -.0274 to -.0285 (ibid p.11). Hirt, Khanna, Ruhnau (EN 43) find short-run elasticity (albeit for potentially shorter timeframes) to be -.05 (ibid. p.4). Pellini (EN 39) reports higher elasticity values, but these values are for long-run elasticity scenarios and, hence, not applicable to the “short-run elasticity” that authors examine.

Overall, authors' poor review and reporting of the pertinent literature raise significant concerns – it would be interesting to see, therefore, how authors' analysis changes under a more realistic elasticity assumption, e.g., in the .05 to .08 range (see data points above).

2. In a similar vein, authors' analysis of their proposed emergency levy's capacity to compensate consumers through recycling of revenues casually omits the reality of administrative and transaction costs associated with such recycling. While authors deserve praise for acknowledging potential behavioral issues with consumer faith in such programs (p.8), they assume that 100% of additional ETS revenues from the emergency levy would find their way into the pockets of ratepayers (see sample calculation for Germany on p.12). Even in the most optimistic scenario, it is unlikely that much over 70% of the emergency levy's revenue, if that much, would actually be disbursed. At present only about 75% of ETS revenues even make it to the member countries who then have to charge their administrative overhead against the remainder.

Other issues:

3. Given the authors' choice of timeframe for their analysis (April 2021 to May 2022), they would do well to expressly acknowledge that the starting point of this period coincided with peak-COVID. Accordingly, the significantly depressed economic activity within the EU during that time actually depressed natural gas prices. As a result, authors are combining two historical crises in their dataset, one (COVID) with a price-depressing, the other (Ukraine) with a price-escalating effect.

4. In their discussion of "Trade-off between gas-to-coal switch and higher electricity price" (p.6), authors would do well to acknowledge that coal is an imperfect substitute for natural gas because coal-fired power plants lack the fast-ramping capability that make natural gas plants such a great complement to intermittent renewables. This mismatch is even more pronounced in the comparison between natural gas and nuclear.

5. The same section further fails to recognize that the ability to switch fuels and dispatch coal-fired power plants in lieu of natural-gas plants is further subject to transmission availability and constraints.

6. On page 3, authors report the state of the TTF as of "the end of 2023" citing to gas futures contracts with delivery in 2026 as "still almost 100 percent higher than pre-crisis level." While these data points might have been current when the manuscript was first written, they are now nearly 2 years out of date. Authors would do well to update their manuscript accordingly, even if more recent data indicates a continuing stabilization of natural gas prices and, thus, a potential weakening of the case for the kind of intervention proposed in the article. See <https://tradingeconomics.com/commodity/eu-natural-gas>.

Language & grammar:

7. Throughout the article, authors use the term "weatherize" to describe how their proposed emergency levy could stabilize natural gas and electricity markets (e.g., p.2, 3, 4). This language is misleading insofar as the crisis scenario (Ukraine crisis and war) that is analyzed was not one prompted by an extreme weather event. "Weatherization" has become a term of art in the electricity sector following events such as the Fukushima Tsunami disaster, the rolling blackouts experienced in the Southern United States during winter storm Uri, etc. The authors would do well to adjust their language accordingly.

8. On page 9, the following sentence should be amended with the italicized insertion: "Given the short time period and the longtime it takes to construct..."

9. On page 11, the following sentence should be revised to include the italicized insertion: "...and thus a 1 EUR increase in carbon price/tax makes coal relatively more expensive..."

I hope these comments are helpful and wish the authors the best of luck with this project!

(Remarks on code availability)

Reviewer #2

(Remarks to the Author)

General remarks:

The study makes an original policy contribution by proposing an emergency levy through carbon prices to stabilize the EU's decarbonization path and mitigate consumer impacts during energy crises. This is a policy-relevant paper, methodologically sound, and adapted to the readership of Nature Communications.

Comments: 1) Econometric robustness: The paper presents carefully designed country-by-country regressions linking coal generation to relative fuel prices and electricity prices to gas prices. Each regression includes hourly, weekday, and monthly fixed effects to account for intra-day and seasonal variations, which is an appropriate way to capture short-term demand and supply patterns. However, the analysis still relies on separate national estimations and does not fully exploit the panel nature of the data or report standard diagnostic tests. Given the hourly frequency of the dataset and the highly volatile context of 2021–2022, additional robustness checks would help ensure that the reported elasticities are statistically reliable and not driven by transient or country-specific effects. To strengthen the empirical credibility of the results, the authors could integrate the following elements:

- Diagnostic tests: report standard tests for autocorrelation and heteroskedasticity, and use corrected (e.g., HAC or clustered) standard errors if needed.

- Cross-country robustness: estimate a pooled model including both country and common time fixed effects to control for EU-wide shocks such as ETS price movements or weather anomalies, and compare the estimated elasticities to the baseline results.

- Dynamic robustness: introduce short lags or moving averages of fuel prices to capture delayed adjustments in generation and electricity prices.

2) Calibration of the emergency levy: The method for determining this “cap-equivalent” carbon price is mentioned as iterative but remains insufficiently detailed.

- The authors should specify the steps used to equate the substitution effects and the sensitivity of this estimate to underlying assumptions.

- Providing a figure or appendix table illustrating this calibration would increase transparency.

2) Interconnections: The analysis treats each country independently, but the European power market is highly integrated. Some discussion (even qualitative) of how electricity

trade and cross-border flows might affect the estimated responsiveness would strengthen the robustness of the results.

3) Long-run considerations: The analysis is restricted to short-term effects. The authors could briefly discuss potential long-term adjustments (such as investment in renewables, storage, or demand flexibility) that might enhance the effectiveness of the proposed mechanism in future crises.

4) Which criteria for activation? The proposed rule-based emergency levy is relevant. However, the paper should clarify possible criteria for activation (deviation from a historical moving average, volatility thresholds?) and discuss the administrative feasibility within the current EU ETS governance.

- The potential “activation rules” must be clearly specified in the Discussion or Methods. The feasibility and the operation design could be discussed in the Discussion.

5) Endogeneity between the CO₂ price and fuel switching: The empirical and counterfactual analyses treat the ETS allowance price as exogenous. However, during the gas crisis, the surge in coal generation itself increased allowance demand and contributed to the rise of the CO₂ price. If the proposed emergency levy had been in place, the resulting lower emissions would have reduced allowance demand, preventing the same upward movement in carbon prices observed during the crisis. Consequently, the paper may overestimate the combined effect of the levy and the existing ETS price level.

- Add a short paragraph clarifying this potential issue of endogeneity and explaining the potential implications.

- If needed, include a robustness check: test whether including EUA prices as an explanatory variable (or instrumenting them) changes the estimated responsiveness.

- A discussion about how feedback between emissions and EUA demand could moderate the policy effect can be appreciated.

6) Feasibility and heterogeneity between European countries: The proposal is ambitious but politically and economically uneven across Member States. Additionally, the proposed emergency levy could, in the short term, destabilize the carbon price signal and create distributional tensions among member states. However, if designed with clear activation rules and predictable redistribution, it could also maybe reinforce the structural credibility of the EU ETS by maintaining a strong carbon signal during crises.

- Discussing these feasibility challenges, including activation criteria, redistribution design, and coordination with the Market Stability Reserve, would enhance the policy realism of the paper.

(Remarks on code availability)

Reviewer #3

(Remarks to the Author)

(Remarks on code availability)

Reviewer #4

(Remarks to the Author)

The paper titled “Emergency Response Mechanisms for addressing challenges with high gas prices in international energy markets” studies how the sharp increase in natural gas prices after the Russian invasion of Ukraine affected electricity generation, CO₂ emissions and wholesale prices across 13 EU countries. Using hourly market data, the authors show substantial substitution from gas to coal and higher emissions during 2021–2022. They then compare the EU’s existing gas price cap to a proposed “emergency levy”, an automatic, rule-based surcharge on the EU ETS allowance price that would

activate when gas prices rise well above historical levels. The idea is that such a levy would keep the relative price between gas and coal constant, thus limiting excess emissions while generating revenues to compensate consumers. The paper mixes credible empirical evidence with a policy proposal meant to make EU climate policy more resilient in energy crises.

The paper tackles an important and timely issue, though the basic insight will be familiar to most economists. It shows, with clear empirical evidence, that gas price caps are a poor way to handle energy price shocks. This is hardly surprising from an economic perspective since price ceilings distort incentives and blunt market adjustment, but it is nonetheless a valuable message in light of the actual policies adopted in Europe during the energy crisis. The authors document some inefficiencies of the cap and show that it is largely ineffective in reducing emissions and has unintended distributional effects. This part needs to be strengthened significantly. It seems to us that there never really was any cap – except in the media. The cap was never invoked and later withdrawn. It is not at all clear who would have paid the difference between an import “market” price and the Ceiling price. This the authors are comparing their policy to something that economic theory and intuition suggests would not work – and that in fact was never used. This is a weakness.

Furthermore, it is strange that the authors do not discuss more the Market Stability Reserve. They say that the EU ETS has no price stabilising mechanism but I think that is just what the MSR is. To date it has mainly been used to deal with the problem of prices being “too low” (and rights are withdrawn). However the MSR could potentially be used to release rights if the prices become too high. This should also be included and discussed.

Intuitively the main part of their argument against the cap on gas prices is that it subsidizes the price leading to an increase in demand and resulting emissions. This is clear and obvious. What is less clear is how their own emergency levy would work. I would like to see a clearer description and analysis of its properties particularly in the context of a tradable permit scheme.

The proposed alternative, the “emergency levy”, is an interesting idea, but it needs significantly better explanation and motivation. In a first-best setting, there would be no need for such a mechanism: the ETS already internalizes the carbon externality, and the emissions cap fixes total emissions irrespective of short-term fuel price movements. The efficient and conceptually clean approach would simply be to let the ETS operate without additional layers. How do we know that the levy does not simply depress the price of permits. The theoretical mechanisms of the levy and interactions between it and the ETS – including the MSR mechanism need to be thought through – or at the very least sketched out.

The paper comes across as missing a vital section describing first how their levy would work and then what exactly the paper does. We go straight from the introduction that criticizes the price cap on gas to “Results” but one is left wondering results of what..

First one would like to see a proper description and analysis of the mechanism proposed, and then of course a presentation of the “model” that leads to the “results”. It appears that there is an econometric analysis of pass through of prices or similar – but this needs to be described and motivated before we get to the results.

A further reflection is that in practice, several second-best constraints might justify their idea. It may not be the economic effects but rather the political effects that are most important. If people think a price cap is needed and that politicians must be seen to be reacting and protecting their electorate then so be it. Large gas price shocks can trigger short-term surges in coal use and emissions that, although later offset within the cap, may appear politically unacceptable and undermine support for carbon pricing. Moreover, during such crises, the carbon price itself may weaken as allowance demand falls, eroding the investment signal for clean technologies (this was not however the case in their Figure 1 Panel D). Perhaps the strongest rationale for the levy might be that it acts as a commitment device: by providing a predictable, rule-based response within the ETS, it constrains policymakers from resorting to ad hoc interventions such as gas price caps that distort incentives and undermine market credibility. In this sense, the proposed levy might perhaps be viewed less as a welfare improvement than as a pragmatic stabilizer, a rule-based device aimed at preserving the political and dynamic integrity of the ETS under extreme market stress.

The political viability of the proposed levy is, however, uncertain. In practice, the levy would raise electricity prices further precisely during periods of already extreme energy costs. This will not help with popularity. The authors place considerable faith in the revenue-refunding mechanism, but it is far from clear that the public would trust or even fully understand such a scheme. For this approach to be politically credible, the redistribution of revenues would have to be made highly transparent and explicit, something that is administratively and politically challenging in practice thus political acceptance is far less than certain.

We also want to add that the EU ETS is already highly complex, with multiple overlapping mechanisms and policy layers on top of the original cap-and-trade design. Adding yet another conditional levy would further increase this complexity and could make the system more difficult to navigate, both for market participants and for policymakers trying to maintain transparency and credibility.

From what we can tell, the empirical work looks solid and convincingly done but it is not properly introduced, explained and contextualized. It should not be the readers job to guess how the levy works and guess how your model estimates are to be read. The analysis might provide interesting insights into how gas price shocks affect coal use, emissions, and wholesale electricity prices across Europe. The estimated relationships seem plausible and consistent with economic reasoning, and the results nicely quantify patterns that are often discussed only qualitatively. We have no major concerns regarding the empirical analysis, except that it needs to be much better placed in context and motivated.

Overall, this is a paper that is uneven. Parts seem well-executed, policy-relevant and interesting but the introduction needs strengthening, the paper lacks a proper description and analysis of the mechanism it proposes and its relation to the ETS

mechanism. Finally the econometric exercise that is the meat of the paper also needs better description and motivation. The study addresses a question of broad scientific and societal relevance and therefore fits well within the scope of Nature Communications. However as it stands now it falls quite far from sufficient standard and we therefore recommend mayor revisions. The revision should clarify the conceptual framing of the emergency levy within a second-best context. In a first-best world, the ETS already delivers the efficient outcome, but in practice political and dynamic constraints may justify temporary corrective mechanisms. The authors should therefore make it clear that their proposal is not meant to perfect the ETS, but to improve outcomes when the first-best logic cannot be sustained.

It would also strengthen the paper to frame the analysis explicitly as a comparison between two second-best strategies, the politically motivated gas price cap and the rule-based emergency levy, rather than as an assessment of an idealized benchmark versus an intervention. Such clarification would sharpen the theoretical logic, align the empirical results with the policy discussion, and make the contribution more transparent to both economists and policymakers.

REFERENCES

See further

Intereconomics / Volumes / 2023 / Number 1 / An EU Price Cap for Natural Gas: A Bad Idea Made Redundant by Market Forces

Volume 58, 2023 · Number 1 · pp. 27–31 · JEL: Q34, Q42, O52
An EU Price Cap for Natural Gas: A Bad Idea Made Redundant by Market Forces
By Daniel Gros

(Remarks on code availability)

Reviewer #5

(Remarks to the Author)

(Remarks on code availability)

Version 1:

Reviewer comments:

Reviewer #1

(Remarks to the Author)

The authors have done a fine job of responding to, and addressing, my comments and criticisms to their earlier manuscript. Most importantly, their clarification of the pertinent literature and the attendant revision of expected short-run elasticities from the previous value of $-.15$ to a more realistic value of $-.06$ (p.13) along with a robustness analysis ranging from $-.03$ to $-.08$ strengthen their analysis and findings.

I also appreciate their express acknowledgment that real-world revenue recycling quotas may differ across EU member states and fall (significantly) short of 100% passthrough. The sample discussion of Spain (pp.5-6) is illuminating and strengthens the case for the proposed mechanism, as does the reference to the EU Social Climate Fund as a possible vehicle to deliver recycled revenues to member states and, eventually, EU citizens (p.8).

Other comments and suggestions of mine have also been addressed and resolved via revisions and/or clarifications to the original manuscript as well as the inclusion of updated data re natural gas futures prices.

Some minor grammatical flaws remain, but these are somewhat natural consequences of the extensive rewriting done by the authors and do not warrant another round of peer reviews before publication. Examples (listed for the authors' benefit) include:

- "We acknowledge some limitations of THE proposed emergency mechanism." (p.8, capitalized article missing)
- "Recent policy debates in Italy—including a reform proposalS to decouple..." (p.6, capitalized letter redundant)

Overall, the authors' diligent revisions have addressed my concerns such that, subject to my fellow reviewers' endorsement of the more quantitative portions of the manuscript, I am now in support of publication.

(Remarks on code availability)

Reviewer #2

(Remarks to the Author)

(Remarks on code availability)

I have just briefly reviewed the code without running it.

Reviewer #3

(Remarks to the Author)

(Remarks on code availability)

Reviewer #4

(Remarks to the Author)

My colleague, with whom i did the original review, and I have both read the revised paper and the answers provided to the referee comments. We are basically satisfied with the revisions and now recommend publication.

(Remarks on code availability)

Reviewer #5

(Remarks to the Author)

(Remarks on code availability)

made.

Response Letter

Responses to Reviewer 1:

This article uses the 2021-22 spike in natural gas prices in Europe, induced by the escalating crisis and eventual war in Ukraine, as a natural experiment to assess the price spike's effect on fuel-shifting from gas to coal along with attendant changes in carbon emissions and wholesale electricity prices. Using hourly electricity market data for select EU countries, the authors model the hypothetical impact that the EU's natural gas price cap (adopted after prices were already normalizing) would have had, if in force at the time. They then compare the results of this modeling exercise to a second hypothetical scenario in which they replace the price cap with an emergency levy added to ETS clearing prices so as to neutralize the price advantage coal-powered electricity generators enjoyed relative to their natural-gas-fueled competitors during the crisis. The authors claim that their proposed emergency levy, assuming revenue recycling via disbursement to ratepayers, would prove superior to the adopted price cap in terms of both lower carbon emissions and lower electricity price burdens for ratepayers.

The article addresses an important question using a relatively novel dataset and methodology; it is well written (for rare errors in grammar see below) and follows a logical structure.

That said, I cannot recommend publication unless and until several issues (some major, others less so) are addressed.

Response: We thank the reviewer for this succinct summary and the helpful remarks on our paper. We have carefully considered all comments below and thoroughly revised our manuscript based on the recommendations of all referees.

Major issues:

1. Much of the claimed superiority of the authors' proposed emergency levy relative to the EU price cap hinges on hypothesized output effects in the form of excess electricity consumption driven by suppressed electricity prices, courtesy of the price cap – compared to overall higher electricity prices in the emergency-levy scenario. But authors reach this conclusion by assuming a degree of price elasticity in electricity demand that the pertinent literature, including the sources cited by the authors themselves, does not support. Specifically, authors assume “an average short-run elasticity of demand of electricity price coherent (sic) with the extant literature of 0.15 (pp.11-12). Yet, the cited literature, including several meta studies, overwhelmingly suggests significantly lower price elasticity values. Cserekylei (EN 40), for example, finds short-run elasticity values of -.08 (residential) and -.10 (industrial) (ibid. p.8). Jin & Kim (EN 37) find even lower short-run elasticity of -.0274 to -.0285 (ibid p.11). Hirt, Khanna, Ruhnau (EN 43) find short-run elasticity (albeit for potentially shorter timeframes) to be -.05 (ibid. p.4). Pellini (EN 39) reports higher elasticity values, but these values are for long-run elasticity scenarios and, hence, not applicable to the “short-run elasticity” that authors examine.

Overall, authors' poor review and reporting of the pertinent literature raise significant concerns – it would be interesting to see, therefore, how authors' analysis changes under a more realistic elasticity assumption, e.g., in the .05 to .08 range (see data points above).

Response: We thank the reviewer for raising this important point regarding the choice of the short-run price elasticity of electricity demand. We have substantially revised the Methods section to clarify both the literature basis and the construction of our baseline parameter.

First, we explicitly distinguish between *short-run* and *long-run* elasticities. Because our analysis focuses on a crisis episode over a limited time horizon—during which capital stocks and appliance portfolios are fixed—we adopt a short-run elasticity.

The literature cited by the reviewer reports short-run elasticities that are considerably smaller than long-run values. For example:

- Csereklyei (2018) reports short-run elasticities of -0.08 for residential consumers and -0.10 for industrial consumers.
- Hirth, Khanna, and Ruhnau (2016) report short-run elasticities around -0.05 .
- Jin and Kim (2021) estimate even smaller short-run elasticities in the range -0.03 .

Long-run elasticities reported in the literature (e.g., Pellini) are substantially larger, but not appropriate for our short-run crisis setting.

To discipline our parameter choice, we construct a sector-weighted short-run elasticity based on the composition of electricity consumption. Specifically, we attempted to use a short run elasticity that reflects the different sectors that demand electricity; we calculate this as a weighted average, where the weights represent the share of electricity consumption by sector—residential, industrial, services, and others. Using EU-average electricity consumption shares (industry $\approx 40\%$, households $\approx 30\%$, services $\approx 25\%$, other $\approx 5\%$) and assigning conservative short-run elasticities consistent with the literature (industrial -0.04 to -0.05 ; residential -0.07 to -0.08 ; services -0.05 ; other -0.03), the implied weighted short-run elasticity lies between -0.05 and -0.06 .

We therefore adopt -0.06 as our central short-run elasticity, which is at the upper end of this weighted range and thus conservative in the sense that it does not understate potential demand responses.

To ensure robustness, we now report sensitivity analyses—with a lower and an upper bound—using elasticities of -0.03 and -0.08 (Supplementary Figure X). While the magnitude of excess demand under the gas price cap naturally varies with the assumed elasticity, the qualitative ranking between the gas price cap and the proposed emergency mechanism remains unchanged across this empirically supported range.

We have updated Figure 5 accordingly and clarified the derivation of the elasticity parameter in the revised Methods section and Supplementary Materials.

2. In a similar vein, authors' analysis of their proposed emergency levy's capacity to compensate consumers through recycling of revenues casually omits the reality of administrative and transaction costs associated with such recycling. While authors deserve praise for acknowledging potential behavioral issues with consumer faith in such programs (p.8), they assume that 100% of additional ETS revenues from the emergency levy would find their way into the pockets of ratepayers (see sample calculation for Germany on p.12). Even in the most optimistic scenario, it is unlikely that much over 70% of the emergency levy's revenue, if that much, would actually be disbursed. At present only about 75% of ETS revenues even make it to the member countries who then have to charge their administrative overhead against the remainder.

Response: We address this important point in the revised Results subsection, “Consumer relief from gas price cap vs. carbon price with revenue recycling.” To help readers assess the real-world capacity of the proposed emergency mechanism, we provide two additional pieces of information. First, Panel D of Figure 5 now reports the share of gross revenues required to deliver the same level of consumer relief as the gas price cap. Second, we explicitly consider a pessimistic scenario in which administrative and transaction costs associated with revenue recycling reduce the funds available for compensation by half. Together, these additions demonstrate that revenue recycling, even net of substantial administrative and transaction costs, would still provide greater consumer relief than the gas price cap in almost all countries. For example, in Spain, only 32% of revenues would be needed to match the relief provided by the gas price cap; even if only 50% of revenues were available for recycling, compensation under the emergency mechanism would remain larger.

In addition, we now discuss the design and challenges of the revenue-refunding mechanism in the two penultimate paragraphs of the revised Discussion. We note that existing EU ETS compensation instruments, such as the Social Climate Fund, provide an institutional basis for redistributing revenues generated by the emergency mechanism to EU Member States according to a transparent and predictable formula, thereby helping to mitigate distributional tensions across heterogeneous EU economies. Member States would then deploy the funds they receive, conditional on EU-approved national Social Climate Plans that specify how citizens are compensated. We argue that it would be in the self-interest of Member States to use the additional revenues generated by the emergency mechanism to provide financial support directly to households during periods of crisis. At the same time, we still acknowledge important political and behavioral considerations, including consumers’ trust in revenue recycling, state capacity to deliver timely transfers, and potential behavioral biases that may lead consumers to overweight electricity price increases while underestimating the benefits of revenue recycling.

Other issues:

3. Given the authors’ choice of timeframe for their analysis (April 2021 to May 2022), they would do well to expressly acknowledge that the starting point of this period coincided with peak-COVID. Accordingly, the significantly depressed economic activity within the EU during that time actually depressed natural gas prices. As a result, authors are combining two historical crises in their dataset, one (COVID) with a price-depressing, the other (Ukraine) with a price-escalating effect.

Response: We now explicitly acknowledge in the revised Method section (third paragraph) that the beginning of our sample period coincides with the late-COVID phase, during which depressed economic activity exerted downward pressure on fossil fuel prices. Absent this demand shock, the subsequent energy price increase, affecting both natural gas and coal, would likely have been even more pronounced. We therefore interpret this issue primarily as a scale effect, which does not affect the qualitative conclusions of our analysis.

4. In their discussion of “Trade-off between gas-to-coal switch and higher electricity price” (p.6), authors would do well to acknowledge that coal is an imperfect substitute for natural gas because coal-fired power plants lack the fast-ramping capability that make natural gas plants such a great complement to intermittent renewables. This mismatch is even more pronounced in the comparison between natural gas and nuclear.

Response: We now discuss that coal is an imperfect substitute for natural gas in the revised section "Trade-off between gas-to-coal switch and higher electricity price."

5. The same section further fails to recognize that the ability to switch fuels and dispatch coal-fired power plants in lieu of natural-gas plants is further subject to transmission availability and constraints.

Response: We fully agree with the reviewer's observation that fuel-switching possibilities are subject to transmission availability, interconnection capacity, and other network constraints.

However, our empirical analysis does not simulate hypothetical dispatch decisions under assumed engineering conditions. Instead, it relies on observed hourly generation and market outcomes during the crisis period. These realized outcomes inherently incorporate all prevailing physical and institutional constraints, including transmission availability, interconnector capacity, congestion, ramping limitations, and market coupling rules. In other words, the estimated responsiveness of coal and gas generation to relative fuel prices reflects substitution that was feasible under actual system constraints at the time. Because we estimate country-specific regressions using realized market data, cross-country differences in elasticities naturally capture heterogeneity in generation mix, grid structure, interconnection capacity, and operational flexibility. Our objective is therefore to quantify the reduced-form effect of relative fuel price movements on dispatch decisions as observed in practice, rather than to construct an engineering counterfactual that abstracts from network constraints. We now clarify this distinction explicitly in the second paragraph of the revised Discussion section.

6. On page 3, authors report the state of the TTF as of "the end of 2023" citing to gas futures contracts with delivery in 2026 as "still almost 100 percent higher than pre-crisis level." While these data points might have been current when the manuscript was first written, they are now nearly 2 years out of date. Authors would do well to update their manuscript accordingly, even if more recent data indicates a continuing stabilization of natural gas prices and, thus, a potential weakening of the case for the kind of intervention proposed in the article. See <https://tradingeconomics.com/commodity/eu-natural-gas>.

Response: We have updated the data, and the revised Figure 1 now includes futures prices through the end of 2025. Notably, in 2025, the average price of gas futures contracts for delivery in 2026 remains nearly 100% above pre-crisis levels.

Language & grammar:

7. Throughout the article, authors use the term "weatherize" to describe how their proposed emergency levy could stabilize natural gas and electricity markets (e.g., p.2, 3, 4). This language is misleading insofar as the crisis scenario (Ukraine crisis and war) that is analyzed was not one prompted by an extreme weather event. "Weatherization" has become a term of art in the electricity sector following events such as the Fukushima Tsunami disaster, the rolling blackouts experienced in the Southern United States during winter storm Uri, etc. The authors would do well to adjust their language accordingly.

Response: We have replaced "weatherize" with "safeguard" or "enhance/increase the resilience," as the original wording could indeed have caused confusion.

8. On page 9, the following sentence should be amended with the italicized insertion: “Given the short time period and the longtime it takes to construct...”

Response: *We changed the sentence.*

9. On page 11, the following sentence should be revised to include the italicized insertion: “...and thus a 1 EUR increase in carbon price/tax makes coal relatively more expensive...”

Response: *We adapted the sentence.*

I hope these comments are helpful and wish the authors the best of luck with this project!

Responses to Reviewers 2 and 3:

General remarks:

The study makes an original policy contribution by proposing an emergency levy through carbon prices to stabilize the EU's decarbonization path and mitigate consumer impacts during energy crises. This is a policy-relevant paper, methodologically sound, and adapted to the readership of Nature Communications.

Comments: 1) Econometric robustness: The paper presents carefully designed country-by-country regressions linking coal generation to relative fuel prices and electricity prices to gas prices. Each regression includes hourly, weekday, and monthly fixed effects to account for intra-day and seasonal variations, which is an appropriate way to capture short-term demand and supply patterns. However, the analysis still relies on separate national estimations and does not fully exploit the panel nature of the data or report standard diagnostic tests. Given the hourly frequency of the dataset and the highly volatile context of 2021–2022, additional robustness checks would help ensure that the reported elasticities are statistically reliable and not driven by transient or country-specific effects. To strengthen the empirical credibility of the results, the authors could integrate the following elements:

Response: We are grateful to the reviewers for their many helpful suggestions. All comments have been carefully addressed, and the manuscript has been thoroughly revised in light of the three referee reports.

- Diagnostic tests: report standard tests for autocorrelation and heteroskedasticity, and use corrected (e.g., HAC or clustered) standard errors if needed.

Response: Our analysis uses clustered standard errors on the day level, as aptly pointed out by the reviewer. This is the appropriate level of clustering given our identification strategy and level of treatment variation. In response to the reviewers' suggestion, we now conducted tests for autocorrelation and heteroskedasticity, specifically with the autocorrelation function (ACF) and partial autocorrelation function (PACF), reported in the new Supplementary Tables 22 and 23. The ACF shows strong intraday persistence and the PACF displays clear dependence at the first few lags. This is consistent with hourly fuel-switching and ramping dynamics one might expect. Day-level clustering thus directly addresses the autocorrelation structure, where standard errors are robust to arbitrary patterns of heteroskedasticity and within-day serial correlation. The diagnostics therefore validate our empirical approach, confirming the day-clustered standard errors are appropriate, which we now highlight in the revised Method section.

- Cross-country robustness: estimate a pooled model including both country and common time fixed effects to control for EU-wide shocks such as ETS price movements or weather anomalies, and compare the estimated elasticities to the baseline results.

Response: We appreciate this suggestion from the reviewers. We now estimate a pooled specification with country fixed effects and common time fixed effects to control for EU-wide shocks like ETS price movements and weather anomalies. The pooled marginal estimate is 0.212 and statistically significant, compared to an average of 0.26 from country-specific estimates. These are qualitatively similar results, suggesting our estimates are not biased by omitted continental-level factors. Fuel-switching appears driven mainly by within-country price variation rather than synchronized EU-wide dynamics. That said, the pooled model

imposes a single elasticity across all countries, which masks real differences in fuel-switching capacity, which our country-specific approach is designed to capture. We report the pooled results in Supplementary Table 21, and discuss them briefly in the Methods section and Supplementary Discussion.

- Dynamic robustness: introduce short lags or moving averages of fuel prices to capture delayed adjustments in generation and electricity prices.

Response: We thank the reviewers for this suggestion. We estimated a specification that augments the baseline with a one-day lag of the relative fuel price alongside the contemporaneous relative price ratio, allowing for potential delayed adjustment in generation decisions. The results are very similar to our specification, where the estimated marginal effect remains virtually identical in magnitude and statistical significance across all countries. Dispatch decisions are governed by current marginal costs, which depend on contemporaneous fuel prices, while any persistence arising from prior unit-commitment decisions plays a secondary role. The robustness check is reported in Supplementary Table 20, and discussed briefly in the Methods section and Supplementary Discussion.

2) Calibration of the emergency levy: The method for determining this “cap-equivalent” carbon price is mentioned as iterative but remains insufficiently detailed. The authors should specify the steps used to equate the substitution effects and the sensitivity of this estimate to underlying assumptions. Providing a figure or appendix table illustrating this calibration would increase transparency.

Response: We appreciate the reviewer’s invitation to clarify the calibration methodology for the equivalent carbon levy. The goal of this exercise is to determine the additional carbon price in 2022 that would yield the same relative price, and thus fuel-switching effect, as the 180 EUR/MWh natural gas price cap would in 2022. This calculation is represented in Equation (1) in the Supplementary Materials and Footnote 5 in the main text. First, we calculate the average relative coal-to-gas price that would prevail under the hypothetical gas cap during 2022. Second, we iteratively solve the equation for the carbon price increment (on top of the actual carbon price) that would produce the same average relative price during 2022. Because coal has a much larger carbon intensity as natural gas, an additional carbon levy disproportionately raises coal’s marginal cost, making gas relatively cheaper—the same directional effect as capping gas prices. Specifically, we test incremental carbon prices from 0.1 to 20.0 EUR/tonne using a fine grid of 10,000 points, calculating the implied relative price at each step until it most closely matches the gas-cap relative price (right-hand side of equation). We find that an additional 12.18 EUR/tonne carbon price would have produced the same coal-to-gas substitution effect during 2022 as the 180 EUR/MWh gas cap. This equivalence holds under our maintained assumptions about standard emissions intensities of coal and natural gas.

2) Interconnections: The analysis treats each country independently, but the European power market is highly integrated. Some discussion (even qualitative) of how electricity trade and cross-border flows might affect the estimated responsiveness would strengthen the robustness of the results.

Response: We fully agree that the European power market is highly integrated and that cross-border electricity trade and interconnector capacity can influence observed dispatch responses.

However, our empirical strategy relies on realized hourly generation and market outcomes within each country. These outcomes reflect equilibrium dispatch under the prevailing market coupling arrangements, cross-border trade, and interconnector constraints during the crisis period. In other words, the estimated fuel-switching elasticities are inclusive of cross-border flows and reflect substitution that was feasible within the integrated European market under actual system conditions.

Because we estimate country-specific regressions, cross-country differences in elasticities naturally capture heterogeneity in generation mix, interconnection capacity, grid structure, and operational flexibility. Our objective is to measure realized price responsiveness under existing institutional and physical constraints, rather than to simulate a counterfactual fully integrated dispatch model.

We now clarify this point explicitly in the second paragraph of the revised Discussion section.

3) Long-run considerations: The analysis is restricted to short-term effects. The authors could briefly discuss potential long-term adjustments (such as investment in renewables, storage, or demand flexibility) that might enhance the effectiveness of the proposed mechanism in future crises.

Response: We thank the reviewers for encouraging us to discuss the longer-run implications of the proposed mechanism.

We agree that our empirical analysis focuses on short-run operational responses and does not explicitly model long-run capital adjustment. In the revised Discussion, we now clarify two complementary longer-run considerations.

First, beyond its short-run emission and consumer effects, the proposed emergency mechanism may generate dynamic benefits by preserving the credibility of the carbon price signal during periods of extreme gas price volatility. By avoiding interventions that suppress electricity prices and weaken carbon incentives, the mechanism may strengthen long-run investment signals for renewables, storage technologies, demand response, and electrification. In contrast, recurrent interventions that suppress electricity prices through gas price caps risk weakening long-run investment signals and increasing policy uncertainty.

Second, we emphasize that the relevance of the mechanism is inherently state-dependent and likely to diminish over time. In the short run, when coal and gas plants remain central to marginal electricity supply, large gas price shocks can induce substantial fuel-switching toward more carbon-intensive generation. However, as decarbonization progresses and investment in renewables, storage, and demand flexibility expands, fossil fuels are less likely to determine the marginal price. In such a setting, extreme gas price movements would have a smaller effect on emissions and wholesale prices, and the emergency mechanism would therefore be activated less frequently. The instrument is thus best understood as a transitional stabilizer whose importance declines as reliance on fossil fuels decreases.

These clarifications have been incorporated in the last paragraph of the revised Discussion section.

4) Which criteria for activation? The proposed rule-based emergency levy is relevant. However, the paper should clarify possible criteria for activation (deviation from a historical moving average, volatility thresholds?) and discuss the administrative feasibility within the current EU ETS governance.

- The potential “activation rules” must be clearly specified in the Discussion or Methods. The feasibility and the operation design could be discussed in the Discussion.

Response: The activation rules and additional details of the operational design are now clearly specified in the seventh paragraph of the revised Discussion. We further clarify that, from an administrative perspective, implementation would be embedded within the existing governance framework of the EU ETS, in particular the Market Stability Reserve and Article 29a, as discussed in the sixth paragraph of the Discussion. The implementation of the emergency mechanism within an extended MSR framework requires defining three rules:

- The first defines the criterion for identifying excessive high natural gas prices that would trigger the mechanism. A simple option is to specify a fixed absolute price level, such as the 180 EUR/MWh threshold used for the EU gas price cap. An alternative option, following the logic of Article 29a, is to define the threshold based on a proportional increase in natural gas prices relative to a historical reference period – for instance, a rise exceeding twice the two-year average natural gas price.
- Upon activation, a second rule determines a reserve price that applies temporarily to all EUA auctions. Under this rule, allowances in auctions are only released when the bid price exceeds the prespecified minimum, what we refer to as the emergency reserve price. A simple way to set this reserve price is to add a fixed surcharge, for example €10 per ton of CO₂, to the average bid price from the preceding auction round. A more flexible alternative, adopted in our policy counterfactual analysis, is to define the top-up based on the actual carbon price increment required to keep the relative price of natural gas to coal constant at a specified historical reference level – for instance, again, the two-year average relative price of natural gas to coal.
- A third rule determines the duration of the emergency activation. Following the design of Article 29a, we propose a fixed period of six consecutive months. After six months, the price situation would be reassessed.

Additional challenges related to feasibility are addressed in the eighth paragraph of the revised Discussion section.

5) Endogeneity between the CO₂ price and fuel switching: The empirical and counterfactual analyses treat the ETS allowance price as exogenous. However, during the gas crisis, the surge in coal generation itself increased allowance demand and contributed to the rise of the CO₂ price. If the proposed emergency levy had been in place, the resulting lower emissions would have reduced allowance demand, preventing the same upward movement in carbon prices observed during the crisis. Consequently, the paper may overestimate the combined effect of the levy and the existing ETS price level.

- If needed, include a robustness check: test whether including EUA prices as an explanatory variable (or instrumenting them) changes the estimated responsiveness.

- A discussion about how feedback between emissions and EUA demand could moderate the policy effect can be appreciated.

Response: We thank the reviewer for raising this important point regarding potential endogeneity between fuel switching and the CO₂ price. To begin with, we clarify that our baseline regressor of interest across all specifications is the relative price of natural gas to coal, *inclusive* of carbon prices. Accordingly, we estimate the fuel switching responsiveness while explicitly controlling for the level of the carbon price. At the same time, we agree that, in principle, increased coal generation during the gas crisis may have raised allowance

demand and contributed to higher EUA prices. However, the available empirical evidence suggests that this feedback channel, which could give rise to endogeneity concerns, is quantitatively limited during periods of market stress and unlikely to materially affect our results.

First, a substantial body of literature finds that fuel switching explains only a small fraction of observed variation in carbon prices (Alberola et al., 2008; Hintermann, 2010; Koch et al., 2014, Mansanet-Bataller et al., 2007). In particular, Koch et al. (2014) show that fuel-switching behavior accounts for less than 10% of variation in EUA prices. Second, the literature emphasizes that EU ETS prices are primarily driven by political and institutional determinants of allowance supply rather than by contemporaneous emissions demand. This reflects the fact that the emissions cap is not static but subject to periodic revisions and unexpected policy interventions. As a result, the literature has long recognized that policymakers governing an ETS face a classic time-inconsistency problem with ex-post incentives to renege on ex-ante commitments (Kydlund and Prescott, 1977; Newell et al., 2014). These pressures intensify during crises, as illustrated by proposals from Poland and other Eastern European countries to suspend or relax the scheduled tightening of the EU ETS cap following the surge in energy prices after the onset of the war in Ukraine. Under such conditions, Salant (2016) and Koch et al. (2016) show that expectations about future policy decisions and supply-side interventions play a dominant role in shaping EUA prices.

Taken together, these findings imply that any feedback from reduced emissions caused by fuel switching to EUA prices is likely to be modest. As a result, our analysis is unlikely to substantially overestimate the combined effect of the emergency levy and the prevailing ETS price level. Nonetheless, we now explicitly clarify this assumption in a new discussion added to the revised Method section (see second paragraph of “Econometric Model of Coal Responsiveness”).

6) Feasibility and heterogeneity between European countries: The proposal is ambitious but politically and economically uneven across Member States. Additionally, the proposed emergency levy could, in the short term, destabilize the carbon price signal and create distributional tensions among member states. However, if designed with clear activation rules and predictable redistribution, it could also maybe reinforce the structural credibility of the EU ETS by maintaining a strong carbon signal during crises.

- Discussing these feasibility challenges, including activation criteria, redistribution design, and coordination with the Market Stability Reserve, would enhance the policy realism of the paper.

Response: First, we address concerns regarding distributional effects and political feasibility in the eighth paragraph of the revised Discussion. Specifically, we note that existing EU ETS compensation instruments, such as the Social Climate Fund, provide an institutional basis for redistributing revenues generated by the emergency mechanism to EU Member States according to a transparent formula. Such an approach is designed to ensure predictability and to mitigate distributional tensions across heterogeneous EU economies.

Second, we acknowledge that the proposed emergency mechanism would add an additional layer to the already complex EU ETS architecture. However, it would build exclusively on existing MSR institutions, making unintended interactions with the current framework unlikely. In particular, as discussed in the penultimate paragraph of the Discussion, a destabilization of the carbon price signal is unlikely. The MSR’s quantity-based rules are

intended to address longer-term structural imbalances in the EU ETS, whereas the proposed price-based emergency mechanism would respond immediately to episodes of excessive gas prices and operate only for a short, predefined period. Given its limited and temporary nature, the emergency intervention is therefore unlikely to materially affect MSR activation.

Responses to Reviewers 4 and 5:

The paper titled “Emergency Response Mechanisms for addressing challenges with high gas prices in international energy markets” studies how the sharp increase in natural gas prices after the Russian invasion of Ukraine affected electricity generation, CO₂ emissions and wholesale prices across 13 EU countries. Using hourly market data, the authors show substantial substitution from gas to coal and higher emissions during 2021–2022. They then compare the EU’s existing gas price cap to a proposed “emergency levy”, an automatic, rule-based surcharge on the EU ETS allowance price that would activate when gas prices rise well above historical levels. The idea is that such a levy would keep the relative price between gas and coal constant, thus limiting excess emissions while generating revenues to compensate consumers. The paper mixes credible empirical evidence with a policy proposal meant to make EU climate policy more resilient in energy crises.

The paper tackles an important and timely issue, though the basic insight will be familiar to most economists. It shows, with clear empirical evidence, that gas price caps are a poor way to handle energy price shocks. This is hardly surprising from an economic perspective since price ceilings distort incentives and blunt market adjustment, but it is nonetheless a valuable message in light of the actual policies adopted in Europe during the energy crisis. The authors document some inefficiencies of the cap and show that it is largely ineffective in reducing emissions and has unintended distributional effects. This part needs to be strengthened significantly. It seems to us that there never really was any cap – except in the media. The cap was never invoked and later withdrawn. It is not at all clear who would have paid the difference between an import “market” price and the Ceiling price. This the authors are comparing their policy to something that economic theory and intuition suggests would not work – and that in fact was never used. This is a weakness.

Response #1: We appreciate the reviewers’ detailed and thoughtful feedback. We have carefully considered each point raised and have substantially revised the manuscript in response.

With respect to the gas price cap, the reviewers are correct that it was never invoked, a point we already stated explicitly in the original submission (see first paragraph of “Effectiveness of EU gas price cap is limited”). Our analysis therefore evaluates a *counterfactual policy*. Using detailed European electricity market data, we assess how the gas price cap would have performed had it been in place during 2022, when natural gas prices reached historically high levels. We agree with the reviewer that the resulting findings, although empirically undocumented to date, are fully consistent with standard economic theory. However, our primary contribution is not to establish this result per se, but rather to use it as a motivation for proposing a novel and more effective emergency mechanism. As detailed below, we have followed the reviewer’s suggestion to clarify the rationale for this “second-best” approach, its implementation within the existing EU ETS governance framework, and the associated challenges for political feasibility.

Furthermore, it is strange that the authors do not discuss more the Market Stability Reserve . They say that the EU ETS has no price stabilising mechanism but I think that is just what the MSR is. To date it has mainly been used to deal with the problem of prices being “too low” (and rights are withdrawn). However the MSR could potentially be used to release rights if the prices become too high. This should also be included and discussed.

Response #2: In the revised Discussion (sixth paragraph), we now explain that the MSR already provides an institutional framework for rule-based interventions, and that extending its mandate offers an administratively feasible way to integrate the proposed emergency mechanism into the existing governance architecture. Specifically, the emergency mechanism could be incorporated through an additional price-based rule: when natural gas prices exceed a prespecified threshold, allowances would be auctioned with a predetermined top-up. We further note that, within the EU ETS, Article 29a already provides for a price-based emergency intervention linked to the carbon price. Under this provision, if EUA prices rise to more than twice their two-year average and the increase is not justified by fundamentals, the European Commission may release additional allowances. However, this mechanism is discretionary and has never been invoked. By contrast, the emergency mechanism proposed here would be rule-based and explicitly linked to natural gas price levels.

Intuitively the main part of their argument against the cap on gas prices is that it subsidizes the price leading to an increase in demand and resulting emissions. This is clear and obvious. What is less clear is how their own emergency levy would work. I would like to see a clearer description and analysis of its properties particularly in the context of a tradable permit scheme.

Response #3: The activation rules and additional implementation details are now clearly specified in the seventh paragraph of the revised Discussion. The implementation of the emergency mechanism within an extended MSR framework requires defining three rules:

- The first defines the criterion for identifying excessive high natural gas prices that would trigger the mechanism. A simple option is to specify a fixed absolute price level, such as the 180 EUR/MWh threshold used for the EU gas price cap. An alternative option, following the logic of Article 29a, is to define the threshold based on a proportional increase in natural gas prices relative to a historical reference period – for instance, a rise exceeding twice the two-year average natural gas price.
- Upon activation, a second rule determines a reserve price that applies temporarily to all EUA auctions. Under this rule, allowances in auctions are only released when the bid price exceeds the prespecified minimum, what we refer to as the emergency reserve price. A simple way to set this reserve price is to add a fixed surcharge, for example €10 per ton of CO₂, to the average bid price from the preceding auction round. A more flexible alternative, adopted in our policy counterfactual analysis, is to define the top-up based on the actual carbon price increment required to keep the relative price of natural gas to coal constant at a specified historical reference level – for instance, again, the two-year average relative price of natural gas to coal.
- A third rule determines the duration of the emergency activation. Following the design of Article 29a, we propose a fixed period of six consecutive months. After six months, the price situation would be reassessed.

The proposed alternative, the “emergency levy”, is an interesting idea, but it needs significantly better explanation and motivation. In a first-best setting, there would be no need for such a mechanism: the ETS already internalizes the carbon externality, and the emissions cap fixes total emissions irrespective of short-term fuel price movements. The efficient and conceptually clean approach would simply be to let the ETS operate without additional layers. How do we know that the levy does not simply depress the price of permits. The theoretical mechanisms of the levy and interactions between it and the ETS – including the MSR mechanism need to be thought through – or at the very least sketched out.

Response #4: First, the proposed rule-based emergency mechanism is now explicitly motivated by political economy considerations as a “second-best” policy in the fifth paragraph of the revised Discussion. We engage in more detail with the reviewer’s comments on the distinction between first-best and second-best settings in response #6 below.

Second, we agree with the reviewer that implementing an emergency levy as a simple top-up to the auction price could depress permit prices, as firms would respond by lowering their bids. In this sense, the levy would act like a tax, reducing willingness to pay and inducing strategic bidding behavior. Accordingly, in the new seventh paragraph of the Discussion on implementation details, we instead propose to implement the emergency mechanism through an auction reserve price, which sets a binding minimum bid and prevents strategic bidding and price depression. Stabilization mechanisms of this type are well established in the literature (Roberts and Spence, 1976; Wood and Jotzo, 2011; Fell et al., 2012) and are already implemented in jurisdictions such as the UK, California, and the U.S. Regional Greenhouse Gas Initiative.

Third, interactions between the proposed emergency mechanism and existing MSR institutions in the EU ETS are now discussed in the penultimate paragraph of the Discussion. We argue that unintended interactions with the current framework are unlikely. The MSR’s quantity-based rules are designed to address longer-term structural imbalances in the EU ETS. Activation is determined by the total number of allowances in circulation, but there is a two-year gap between the year it is measured and the year of the MSR intake or release. By contrast, the proposed price-based emergency mechanism would respond immediately to episodes of excessive gas prices and operate only for a short, predefined period (e.g., six months). Given this limited and temporary nature, it is therefore unlikely that the emergency intervention would materially affect MSR activation two years later.

The paper comes across as missing a vital section describing first how their levy would work and then what exactly the paper does. We go straight from the introduction that criticizes the price cap on gas to “Results” but one is left wondering results of what..

First one would like to see a proper description and analysis of the mechanism proposed, and then of course a presentation of the “model” that leads to the “results”. It appears that there is an econometric analysis of pass through of prices or similar – but this needs to be described and motivated before we get to the results.

Response #5: The manuscript is structured in line with the standard format of Nature Communications, in which the Results section follows directly after the Introduction, while detailed methodological descriptions are presented in a dedicated Methods section at the end of the paper. This structure reflects the journal’s emphasis on presenting the main findings and their interpretation upfront, with technical and econometric details provided subsequently.

Within this format, a detailed discussion of our policy proposal, which follows from the empirical findings presented in the Results section, is necessarily confined to the Discussion. We acknowledge that, in the original submission, the description of the proposed emergency levy in this section was not sufficiently clear or well structured. In response, we have substantially revised the Discussion to provide a clear and systematic explanation of how the emergency levy would operate in practice and how it would integrate with existing EU ETS institutions.

In addition, we have revised the Methods section at the end of the paper, as well as the figure captions, to make it easier for readers to connect the presented results to the underlying econometric analyses (one estimating price elasticities and the other estimating pass-through rates) as well as to the counterfactual analysis.

A further reflection is that in practice, several second-best constraints might justify their idea. It may not be the economic effects but rather the political effects that are most important. If people think a price cap is needed and that politicians must be seen to be reacting and protecting their electorate then so be it. Large gas price shocks can trigger short-term surges in coal use and emissions that, although later offset within the cap, may appear politically unacceptable and undermine support for carbon pricing. Moreover, during such crises, the carbon price itself may weaken as allowance demand falls, eroding the investment signal for clean technologies (this was not however the case in their Figure 1 Panel D). Perhaps the strongest rationale for the levy might be that it acts as a commitment device: by providing a predictable, rule-based response within the ETS, it constrains policymakers from resorting to ad hoc interventions such as gas price caps that distort incentives and undermine market credibility. In this sense, the proposed levy might perhaps be viewed less as a welfare improvement than as a pragmatic stabilizer, a rule-based device aimed at preserving the political and dynamic integrity of the ETS under extreme market stress.

Response #6: We thank the reviewers for this important comment. As noted above, in the fifth paragraph of the revised Discussion we now motivate the proposed emergency mechanism as a “second-best” policy. We begin by acknowledging that, in a first-best setting, temporary increases in coal use and emissions would not be problematic, as cumulative emissions are fixed by the EU ETS cap. However, following the reviewers’ suggestion, we argue that this perspective overlooks two key political-economy dynamics that are highly relevant in practice:

1. High energy prices tend to trigger policy debates about the political commitments to the EU ETS cap. Because the cap is not static but subject to periodic updates and unexpected policy revisions, the literature has long recognized that policymakers governing an ETS face a classic time-inconsistency problem with ex-post incentives to renege on ex-ante commitments (Kydland/Prescott 1977, Newell et al. 2014) – with these pressures intensifying during periods of crisis (Koch et al. 2016, Salant 2016). The proposed emergency mechanism is therefore conceived as a commitment device designed to preserve the political and dynamic integrity of the ETS under conditions of market stress.
2. A first-best perspective abstracts from policymakers’ concerns about the political repercussions of distributional impacts associated with high energy prices during crises. Such concerns frequently give rise to calls for relaxing climate policy or for alternative interventions that promise immediate financial relief. In the absence of a readily available, rule-based mechanism to address these pressures, policymakers often resort to ad hoc measures that distort incentives and are limited in effectiveness. The EU’s introduction of a gas price cap in 2022 provides a salient example. By contrast, the proposed emergency mechanism would offer a predictable, rule-based response to distributional concerns, thereby constraining policymakers from resorting to ad hoc interventions.

The political viability of the proposed levy is, however, uncertain. In practice, the levy would raise electricity prices further precisely during periods of already extreme energy costs. This

will not help with popularity. The authors place considerable faith in the revenue-refunding mechanism, but it is far from clear that the public would trust or even fully understand such a scheme. For this approach to be politically credible, the redistribution of revenues would have to be made highly transparent and explicit, something that is administratively and politically challenging in practice thus political acceptance is far less than certain.

Response #7: The reviewers again raise an important concern that warrants further attention. In response, we now discuss distributional considerations, as well as the design and potential challenges of the revenue-refunding mechanism, in the two penultimate paragraphs of the revised Discussion.

Specifically, we note that existing EU ETS compensation instruments, such as the Social Climate Fund, provide an institutional basis for redistributing revenues generated by the emergency mechanism to EU Member States according to a transparent and predictable formula, thereby helping to mitigate distributional tensions across heterogeneous EU economies. Member States would then deploy the funds they receive, conditional on EU-approved national Social Climate Plans that specify how citizens are compensated. We argue that it would be in the self-interest of Member States to use the additional revenues generated by the emergency mechanism to provide financial support directly to households during periods of crisis. At the same time, we acknowledge important political and behavioral considerations, including consumers' trust in revenue recycling, state capacity to deliver timely transfers, and potential behavioral biases that may lead consumers to overweight electricity price increases while underestimating the benefits of recycled revenues.

We also want to add that the EU ETS is already highly complex, with multiple overlapping mechanisms and policy layers on top of the original cap-and-trade design. Adding yet another conditional levy would further increase this complexity and could make the system more difficult to navigate, both for market participants and for policymakers trying to maintain transparency and credibility.

Response #8: We now acknowledge that the proposed emergency mechanism would add an additional layer to the already complex EU ETS architecture (see the penultimate paragraph of the revised Discussion). However, as noted in response #4 above, it builds exclusively on existing MSR institutions, making unintended interactions with the current framework unlikely.

From what we can tell, the empirical work looks solid and convincingly done but it is not properly introduced, explained and contextualized. It should not be the readers job to guess how the levy works and guess how your model estimates are to be read. The analysis might provide interesting insights into how gas price shocks affect coal use, emissions, and wholesale electricity prices across Europe. The estimated relationships seem plausible and consistent with economic reasoning, and the results nicely quantify patterns that are often discussed only qualitatively. We have no major concerns regarding the empirical analysis, except that it needs to be much better placed in context and motivated.

Overall, this is a paper that is uneven. Parts seem well-executed, policy-relevant and interesting but the introduction needs strengthening, the paper lacks a proper description and analysis of the mechanism it proposes and its relation to the ETS mechanism. Finally the econometric exercise that is the meat of the paper also needs better description and motivation. The study addresses a question of broad scientific and societal relevance and therefore fits well within the scope of Nature Communications. However as it stands now it falls quite far from sufficient standard and we therefore recommend mayor revisions. The

revision should clarify the conceptual framing of the emergency levy within a second-best context. In a first-best world, the ETS already delivers the efficient outcome, but in practice political and dynamic constraints may justify temporary corrective mechanisms. The authors should therefore make it clear that their proposal is not meant to perfect the ETS, but to improve outcomes when the first-best logic cannot be sustained.

It would also strengthen the paper to frame the analysis explicitly as a comparison between two second-best strategies, the politically motivated gas price cap and the rule-based emergency levy, rather than as an assessment of an idealized benchmark versus an intervention. Such clarification would sharpen the theoretical logic, align the empirical results with the policy discussion, and make the contribution more transparent to both economists and policymakers.

Response #9: We again thank the reviewers for the helpful suggestions. We have engaged with all comments summarized here in our responses above. We hope that the revised manuscript overcomes many of the limitations that the reviewers accurately pointed out.

REFERENCES

See further

Intereconomics / Volumes / 2023 / Number 1 / An EU Price Cap for Natural Gas: A Bad Idea Made Redundant by Market Forces

Volume 58, 2023 · Number 1 · pp. 27–31 · JEL: Q34, Q42, O52

An EU Price Cap for Natural Gas: A Bad Idea Made Redundant by Market Forces

By Daniel Gros

We cite the suggested reference in the revised manuscript.

Response Letter

Responses to Reviewer 1:

The authors have done a fine job of responding to, and addressing, my comments and criticisms to their earlier manuscript.

Most importantly, their clarification of the pertinent literature and the attendant revision of expected short-run elasticities from the previous value of $-.15$ to a more realistic value of $-.06$ (p.13) along with a robustness analysis ranging from $-.03$ to $-.08$ strengthen their analysis and findings.

I also appreciate their express acknowledgment that real-world revenue recycling quotas may differ across EU member states and fall (significantly) short of 100% passthrough. The sample discussion of Spain (pp.5-6) is illuminating and strengthens the case for the proposed mechanism, as does the reference to the EU Social Climate Fund as a possible vehicle to deliver recycled revenues to member states and, eventually, EU citizens (p.8).

Other comments and suggestions of mine have also been addressed and resolved via revisions and/or clarifications to the original manuscript as well as the inclusion of updated data re natural gas futures prices.

Response: We thank the reviewer for this positive feedback.

Some minor grammatical flaws remain, but these are somewhat natural consequences of the extensive rewriting done by the authors and do not warrant another round of peer reviews before publication. Examples (listed for the authors' benefit) include:

- "We acknowledge some limitations of THE proposed emergency mechanism." (p.8, capitalized article missing)
- "Recent policy debates in Italy—including a reform proposals to decouple..." (p.6, capitalized letter redundant)

Response: We have carefully reviewed the manuscript and corrected the grammatical issues.

Overall, the authors' diligent revisions have addressed my concerns such that, subject to my fellow reviewers' endorsement of the more quantitative portions of the manuscript, I am now in support of publication.